# Feedback regulation of cytoneme-mediated transport shapes a tissue-specific FGF morphogen gradient

**Lijuan Du, Alex Sohr, Ge Yan, Sougata Roy\***

Department of Cell Biology and Molecular Genetics, University of Maryland, Maryland, United States

**Abstract** Gradients of signaling proteins are essential for inducing tissue morphogenesis. However, mechanisms of gradient formation remain controversial. Here we characterized the distribution of fluorescently-tagged signaling proteins, FGF and FGFR, expressed at physiological levels from the genomic knock-in alleles in *Drosophila*. FGF produced in the larval wing imaginal-disc moves to the air-sac-primordium (ASP) through FGFR-containing cytonemes that extend from the ASP to contact the wing-disc source. The number of FGF-receiving cytonemes extended by ASP cells decreases gradually with increasing distance from the source, generating a recipient-specific FGF gradient. Acting as a morphogen in the ASP, FGF activates concentration-dependent gene expression, inducing *pointed-P1* at higher and *cut* at lower levels. The transcription-factors Pointed-P1 and Cut antagonize each other and differentially regulate formation of FGFR-containing cytonemes, creating regions with higher-to-lower numbers of FGF-receiving cytonemes. These results reveal a robust mechanism where morphogens self-generate precise tissue-specific gradient contours through feedback regulation of cytoneme-mediated dispersion.

DOI: https://doi.org/10.7554/eLife.38137.001

**\*For correspondence:** sougata@umd.edu

**Competing interests:** The authors declare that no competing interests exist.

## Introduction

Diverse tissue shapes and patterns are created by conserved paracrine signaling proteins such as Transforming Growth Factor β (TGFβ), Hedgehog (Hh), Wingless (Wg/WNT), Epidermal Growth Factor (EGF) and Fibroblast growth factor (FGF). Irrespective of the diversity of the functions and forms created, a universal mode by which these signals coordinate responses in a large population of cells involves the generation of concentration gradients. For instance, morphogen gradients induce concentration-dependent gene activities that regulate tissue growth and patterning (reviewed in (*Bier and De Robertis, 2015*; *Christian, 2012*; *Restrepo et al., 2014*; *Rogers and Schier, 2011*; *Wolpert, 2016*)). Similarly, growth factor and chemokine gradients are thought to control cell migration during vasculogenesis, neurogenesis, wound healing, and immune responses (*Cai and Montell, 2014*; *Majumdar et al., 2014*). Despite advances in our understanding of signal transduction pathways, how signals disperse and how the dispersion mechanism is dynamically modulated to shape gradients in three-dimensional tissue structures are poorly understood. Moreover, the formation of signal gradients is unexplored in most morphogenetic contexts, so we do not know how dispersion of the signals through extracellular space can generate the required diversity in gradient shapes and contours for a multitude of tissue architectures.

Several models primarily based on free/restricted/facilitated extracellular diffusion of signals were proposed previously to explain the establishment of signal gradients, but the proposed mechanisms are highly controversial (*Majumdar et al., 2014*; *Müller et al., 2013*; *Wolpert, 2016*). Recent advancements in live imaging technology facilitated the discovery of a radically different mechanism of signal dispersion, where signals move directly from producing to recipient cells through actin-

**eLife digest** When an embryo develops, its cells must work together and 'talk' with each other so they can build the tissues and organs of the body. A cell can communicate with its neighbors by producing a signal, also known as a morphogen, which will tell the receiving cells what to do. Once outside the cell, a morphogen spreads through the surrounding tissue and forms a gradient: there is more of the molecule closer to the signaling cells and less further away. The cells that receive the message respond differently depending on how much morphogen they get, and therefore on where they are placed in the embryo.

How morphogens move in tissues to create gradients is still poorly understood. One hypothesis is that, once released, they spread passively through the space between cells. Instead, recent research has shown that some morphogens travel through long, thin cellular extensions known as cytonemes. These structures directly connect the cells that produce a morphogen with the ones that receive the molecule. Yet, it is still unclear how cytonemes can help to form gradients.

Du et al. aimed to resolve this question by following a morphogen called Branchless as it traveled through fruit fly embryos. Branchless is important for sculpting the embryonic airway tissue into a delicate network of branched tubes which supply oxygen to the cells of an adult fly. However, no one knew how cells communicate Branchless, whether or not Branchless formed a gradient, and if it did, how this gradient was created to set up the plan to form airway tubes. It was assumed that the molecule would diffuse passively to reach airway cells – but this is not what the experiments by Du et al. showed.

To directly observe how Branchless moves among cells, insects were genetically engineered to produce Branchless molecules attached to a fluorescent 'tag'. Microscopy experiments using these flies revealed that Branchless did not diffuse passively; instead, airway cells used cytonemes to 'reach' towards the cells that produced the molecule, collecting the signal directly from its source. The gradient was created because the airway cells near the cells that make Branchless had more cytonemes, and therefore received more of the molecule compared to the cells that were placed further away. Genetic analysis of the airway tissue showed that Branchless acts as a morphogen to switch on different genes in the receiving cells placed in different locations. The target genes activated by the gradient instruct the receiving cells on how many cytonemes need to be extended, which helps the gradient to maintain itself over time.

Du et al. demonstrate for the first time how cytonemes can relay a signal to establish a gradient in a developing tissue. Dissecting how cells exchange information to create an organism could help to understand how this communication fails and leads to disorders.

DOI: https://doi.org/10.7554/eLife.38137.002

based specialized filopodia or cytonemes (*Bischoff et al., 2013*; *Eom and Parichy, 2017*; *Roy et al., 2011b*; *Roy et al., 2014*; *Sanders et al., 2013*; *Stanganello and Scholpp, 2016*). Cytonemes and cytoneme-like projections are now known to be required for cell-cell communication mediated by many families of signaling proteins (*Kornberg, 2014*). For instance, a cytoneme-dependent mechanism was shown to be essential for generating concentration-dependent signaling responses to the Hh and Dpp morphogens (*Bischoff et al., 2013*; *Chen et al., 2017*; *Roy et al., 2014*). However, how local cytoneme-mediated cell-to-cell communication might create long-range signal gradients to regulate tissue morphogenesis was unknown.

In this study, we aimed to address these fundamental questions by characterizing gradient formation of a *Drosophila* FGF family protein, Branchless (Bnl). In *Drosophila*, Bnl is the primary signal that guides the branching morphogenesis of tracheal epithelial tubes (*Sutherland et al., 1996*). At the third instar larval stage, Bnl produced from a restricted group of wing imaginal disc cells induces budding and growth of a wing disc-associated tracheal branch, the air-sac primordium (ASP), from the disc-associated transverse connective (TC) (*Figure 1A*) (*Sato and Kornberg, 2002*). ASP tracheoblast cells are precursors of adult air-sac, an organ that is analogous to the vertebrate lung. Traditionally, all paracrine signals such as FGF family proteins are thought to function as diffusible signals and form a gradient by passive extracellular dispersion. In accordance with the same line of thinking, one hypothesis predicts that *Drosophila* Bnl diffuses from its source to form an extracellular gradient

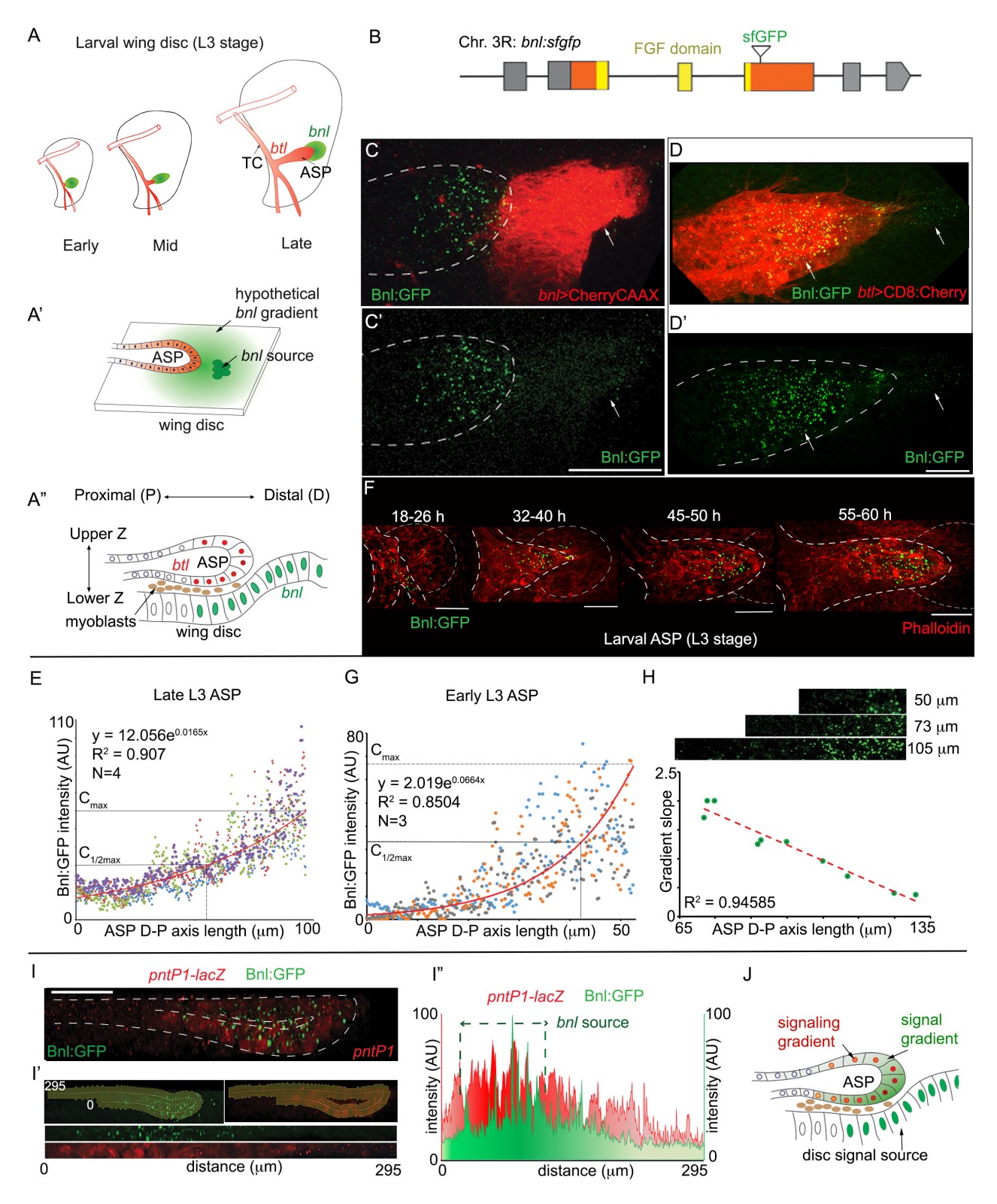

**Figure 1.** A concentration gradient of Bnl:GFP adopts precise morphologies of the recipient ASP. (**A**) Drawings depicting budding and directed growth of the third instar larval ASP (red, *btl* expression) regulated by Bnl produced in a restricted group of cells in the wing disc (green); Bnl source spatiotemporally changes position ahead of the growing ASP; TC, transverse connective. (**A'**) Drawing depicting hypothetical chemotactic gradient of secreted Bnl (green) that was predicted to guide the directed ASP (red) growth toward the Bnl expressing cells (green circle). (**A''**) Drawing depicting a

*Figure 1 continued on next page*

*Figure 1 continued*

cross-section of the late third instar larval ASP and wing disc, showing their epithelial contours, relative position in X-Z-Y dimension, putative Bnl-responsive cells (red), and disc *bnl*-expressing cells (green); upper/lower Z. (B) A schematic map of *bnl:gfp$^{endo}$* knock-in allele; grey box, non-coding exons; orange box, coding exons; line, introns. (C–D') Z-projected images showing that Bnl:GFP, produced at physiological level from the *bnl:gfp$^{endo}$* allele, moved from the disc *bnl*-source to the ASP and distributed along the distal to proximal direction of the recipient tissue; (C,C') *bnl*-source marked by CherryCAAX expression (*bnl-LexA, lexO-CherryCAAX/bnl:gfp$^{endo}$*); (D,D') recipient ASP marked by CD8:Cherry expression (*btl-Gal4,UAS-CD8: Cherry/+; bnl:gfp$^{endo}$*); (C,D) merged red and green channels; (C',D') only the green channel; arrows, Bnl:GFP signal detected specifically in the wing disc source and recipient ASP. (E) Graph showing the Bnl:GFP concentration gradient along the ASP D-P axis in late third instar larvae (N = 4 independent samples). (F) Coordination of Bnl:GFP gradient formation with the ASP growth; time points, hours (h) after third instar larval molt; relative position of *bnl*-source marked by dashed-line. (G) Narrow range of Bnl:GFP gradient in shorter ASPs from early third instar larvae (N = 3 independent samples). (E,G) Red line graph, trend-line of the X-Y scatter plot with exponential fit from the averaged value; C$_{max}$, maximum average Y value (Bnl:GFP intensity); (H) Negative correlation of the ASP D-P axis length and the slope (C$_{max}$ to C$_{1/2max}$) of the Bnl:GFP gradient; each coordinate represents a single disc-ASP tissue; upper panels, Bnl:GFP distribution in three examples of ASPs with different lengths of D-P axis (the cropped region). (I) A 3D sagittal view showing a continuous long-range Bnl:GFP distribution across the entire recipient ASP epithelium adopting its tubular contour; expression of a target gene reporter of Bnl signaling, *pntP1-lacZ* (red, anti-βGal) showing corresponding signaling response; (I',I'') Intensity plots of Bnl:GFP (green) and *pntP1-lacZ* (red) (I'') across the entire ASP epithelium derived from the digitally straightening ASP epithelium shown in I'. (J) Drawing of a cross section of the ASP-wing disc, summarizing the observations from C,D,I. (C–I') Fixed samples, Z-projection, except I-I'; AU, arbitrary unit; dashed line, ASP or wing disc outline. Scale bars, 30 μm.

DOI: https://doi.org/10.7554/eLife.38137.003

The following source data and figure supplements are available for figure 1:

**Source data 1.** Data for the intensity profile plot of Bnl:GFP along the D-P axis of the late 3$^{rd}$ instar larval ASP in *Figure 1E*.
DOI: https://doi.org/10.7554/eLife.38137.006
**Source data 2.** Data for the intensity profile plot of Bnl:GFP along the D-P axis of the early 3$^{rd}$ instar larval ASP in *Figure 1G*.
DOI: https://doi.org/10.7554/eLife.38137.007
**Source data 3.** Numerical data for correlation of the length of the D-P axis of the ASP and the slope of the Bnl:GFP gradient in *Figure 1H*.
DOI: https://doi.org/10.7554/eLife.38137.008
**Source data 4.** Data for the intensity profile plots of Bnl:GFP and *pntP1-lacZ* along the digitally straightened tubular ASP epithelium in *Figure 1I"*, *Figure 1—figure supplement 2D"*, and *Figure 1—figure supplement 2E"*.
DOI: https://doi.org/10.7554/eLife.38137.009
**Figure supplement 1.** Generating genome-edited flies harboring a *bnl:gfp$^{endo}$* allele.
DOI: https://doi.org/10.7554/eLife.38137.004
**Figure supplement 2.** Characterization of Bnl:GFP distribution in the ASP.
DOI: https://doi.org/10.7554/eLife.38137.005

and that the gradient guides directional migration of the tracheal branches such as the ASP (*Horowitz and Simons, 2008*; *Ochoa-Espinosa and Affolter, 2012*) (*Figure 1A'*). However, whether Bnl forms a gradient and if so, how Bnl is transported and modulated in the extracellular space to form a gradient remain unexplored. Moreover, all tracheal cells express the Bnl receptor, Breathless (Btl), and develop under the guidance of the same Bnl signal, but different embryonic and larval tracheal branches adopt different developmental stage-specific morphologies (*Sato and Kornberg, 2002*; *Sutherland et al., 1996*). We do not understand whether and how the shape of a Bnl gradient can dynamically adapt to diverse tissue morphologies.

To investigate these important gaps in knowledge, we chose to focus on the *Drosophila* larval ASP because of its unique features. In this system, the wing disc *bnl*-source is spatially separated from the recipient ASP epithelium (*Figure 1A"*). Bnl signals exclusively to the ASP because the wing disc cells and myoblasts located between the disc and the ASP do not express its receptor, Btl. Moreover, the two additional *Drosophila* FGF family proteins, Pyramus and Thisbe, do not share receptors with Bnl (*Sato and Kornberg, 2002*; *Stathopoulos et al., 2004*). This system where there is a clear inter-organ communication path and a single Bnl-specific receptor exclusively expressed in the ASP is ideal for an unbiased interpretation of the extracellular route of Bnl transport and gradient formation. Genome-edited Bnl:GFP and Btl:Cherry constructs generated in this study revealed Bnl distribution with high sensitivity and precision, allowing direct visualization of an endogenous signal and signaling gradient. We showed that Bnl moves target-specifically from the producing cells to the ASP through cytonemes and forms a long-range concentration gradient by dynamically adopting the recipient ASP-specific shapes and contours. We also demonstrated that Bnl functions as a morphogen and, most importantly, we uncovered a novel self-regulatory mechanism of cytoneme-

mediated signaling by which the Bnl morphogen gradient forms. Thus, these findings showed an example of morphogenesis in which cytoneme-dependent signaling can provide precision and adaptability in shaping long-range positional gradients and tissue architectures in space and time.

## Results

### A concentration gradient of Bnl:GFP adopts the recipient ASP-specific contour

To visualize Bnl dispersion without affecting normal tracheal morphogenesis, we employed the CRISPR/Cas9-based genome-editing technique to generate flies that harbor an in-frame insertion of a superfolder-GFP (sfGFP) sequence within the third coding exon of the *bnl* gene (*Figure 1B*; *Figure 1—figure supplement 1A–C*; Materials and methods). Animals that harbored the *bnl:gfp^endo* knock-in allele were homozygous viable, had normal tissue morphology, and the expected gene expression patterns (*Du et al., 2017*; *Sato and Kornberg, 2002*). Under sensitive confocal microscopy (see Materials and methods), endogenous Bnl:GFP molecules were visualized as fluorescent puncta (*Figure 1C,C'*). The distribution patterns of these Bnl:GFP puncta revealed several unexpected features. In the wing disc source cells that were marked by CherryCAAX expression, Bnl:GFP puncta were faint, hardly detectable, and homogeneously distributed (*Figure 1C–D'*). As a paracrine signal, Bnl was expected to disperse and form a continuous gradient surrounding the wing disc source (*Figure 1A'*). However, all of the detectable bright fluorescent Bnl:GFP puncta outside of the source cells were asymmetrically distributed only in the recipient ASP cells (*Figure 1C–D'*). Thus, although Bnl:GFP moved from the wing disc source to the ASP, the punctate signal did not localize in the non-specific wing disc cells surrounding the signal source or in the myoblast cells located between the disc source and the ASP (*Videos 1* and *2*).

To ensure that the GFP-marked puncta in the ASP were actual Bnl molecules, we performed an immunohistochemistry assay (IHC) with αBnl antibody on the homozygous *bnl:gfp^endo* larval tissues. The Bnl antibody recognized all of the Bnl:GFP puncta in the larval ASP, confirming that they represent the Bnl protein (*Figure 1—figure supplement 2A,A'*). To further validate whether the Bnl:GFP distribution in the ASP represented a functional distribution of the ligand, we knocked-down *bnl:gfp^endo* expression in the wing disc source. Since Bnl is essential for tracheal/ASP growth (*Sutherland et al., 1996*; *Sato and Kornberg, 2002*), loss of a functional Bnl:GFP distribution was expected to abrogate ASP development. As expected, overexpression of *bnlRNAi* from the disc *bnl*-source in the *bnl:gfp^endo* larvae knocked down both Bnl:GFP expression and ASP growth in most samples, indicating that the tagged signaling protein is functional (*Figure 1—figure supplement 2B–C*). However, due to incomplete *RNAi*-mediated knock-down of *bnl:gfp^endo*, a few samples still contained low levels of Bnl:GFP. Importantly, these samples also had small, stunted ASPs with a shallow range of Bnl:GFP distribution. This correlation between Bnl:GFP levels/distribution and ASP growth suggested that the levels and distribution of Bnl:GFP in the ASP have a significant developmental role (*Figure 1—figure supplement 2B–C*).

Contrary to what was predicted earlier (*Figure 1A'*; (*Horowitz and Simons, 2008*; *Ochoa-Espinosa and Affolter, 2012*)), our imaging results (*Figure 1C–D'*) suggested that a pre-existing Bnl:GFP gradient surrounding the wing disc signal source was not formed. Instead, the

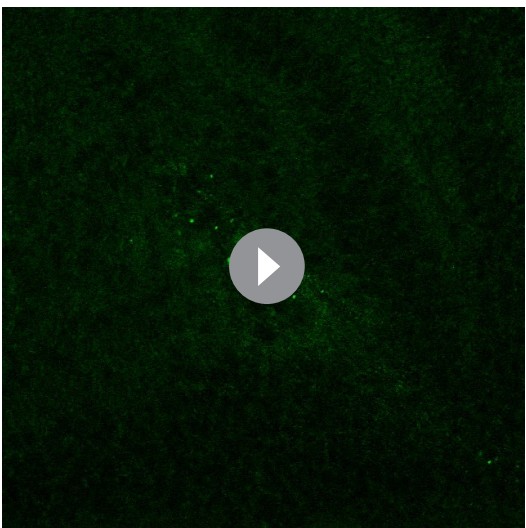

**Video 1.** Asymmetric distribution of Bnl:GFP produced from the wing disc source. Serial Z-stacks of the CD8:Cherry-marked ASP and underlying wing disc Bnl:GFP producing cells shown from lower to upper optical sections; only the green channel is shown to highlight asymmetric Bnl:GFP localization; genotype: *btl-Gal4, UAS-CD8:Cherry/+; bnl:gfp^endo*.
DOI: https://doi.org/10.7554/eLife.38137.010

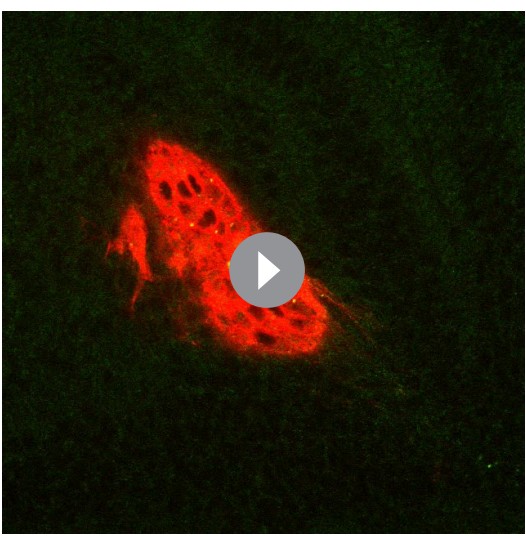

**Video 2.** A recipient ASP-specific Bnl:GFP distribution. Same sample and Z-stacks of the CD8:Cherry-marked ASP as shown in **Video 1**, but now with both red and green channels to highlight that the asymmetric Bnl: GFP distribution was due to dispersion of Bnl:GFP only in the recipient ASP epithelium; genotype: *btl-Gal4, UAS-CD8:Cherry/+; bnl:gfp^endo*.
DOI: https://doi.org/10.7554/eLife.38137.011

Bnl:GFP puncta appeared to distribute in a graded pattern along the distal-to-proximal (D-P) axis of the recipient ASP (**Figure 1D,D'**). Measurement of Bnl:GFP intensities along the longest D-P axis of the late third instar larval ASPs revealed a long-range exponential gradient spanning ~100–120 μm in range. The distal ASP tip, which is nearest to the disc *bnl*-source, received the highest levels of Bnl:GFP (**Figure 1E**). The signal concentration decreased exponentially toward the proximal ASP stalk as the distance from the source increased. Thus, although Bnl:GFP is produced in the wing disc, its gradient is formed in the recipient ASP.

The Bnl:GFP gradient shapes were variable, and developed concomitantly with the growth of the recipient ASP (**Figure 1F**). In the early third instar larval stage (18–26 hr after third instar larval molt), the ASP was hardly detectable. At this stage, only a few TC cells proximal to the *bnl*-source had received Bnl:GFP. In the later stages, as the ASP budded from the TC and then elongated, the Bnl:GFP gradient shapes adopted the variable shapes of the recipient ASP. These results suggested that the shape of the Bnl:GFP gradient is adaptable and its formation is coordinated with growth of the recipient ASP. We further examined the relationship of the slope of the Bnl:GFP gradient with the length of the D-P axis of the recipient ASP. In the shorter early third instar larval ASPs (~35–40 hr after third instar larval molt), Bnl:GFP formed relatively steeper and narrower gradients (slope of a straight line joining $C_{max}$ to $C_{1/2max}$ =~3.14) compared to the larger/longer late-stage ASPs (slope =~0.71) (**Figure 1E, G**). A strong negative correlation of the gradient slopes with the length of the ASP D-P axis was also observed in a population of randomly harvested wandering third instar larval ASPs, which showed natural developmental variations in size (**Figure 1H**). These results showed that the range and slope of the Bnl:GFP gradients are dependent on the ASP morphology and that the gradient did not preexist, but instead developed dynamically in coordination with the recipient tissue morphogenesis.

Although 2D Z-projected images were sufficient to visualize the gradient in the X-Y plane, the ASP epithelium has a complex three-dimensional tubular morphology. As shown in the cross-section diagram of the ASP and wing disc in **Figure 1A"**, the relative distance of the ASP cells from the disc source can be measured along the D-P axis as well as in the upper and lower layers across the vertical Z-axis. Across the Z-axis, most of the cells in the upper ASP layer (ventral face) were situated away from the underlying *bnl*-source. In spite of the physical distances, Bnl:GFP distribution was not restricted to the ASP cells that were immediately juxtaposed to the *bnl*-source. Instead, Bnl:GFP distributed in both layers formed a continuous long-range gradient, adopting the complex 3D tubular contour of the epithelium (~300 μm long, equivalent to 30 cells in just a single medial cross-section) (**Figure 1I–I"**). When we probed for the expression of a Bnl-signaling target gene *pntP1*, an ETS family transcription factor (**Sutherland et al., 1996**), overlapping signal and signaling gradients were revealed (**Figure 1I–J**, **Figure 1—figure supplement 2D–E"**). Interestingly, in a digitally straightened continuous epithelium, the signal and signaling gradients had a bilateral symmetric distribution, where the positions with highest concentrations corresponded to the ASP cells nearest to the disc *bnl*-source (**Figure 1I',I",J**). Tubular contours of the ASP epithelium were variable among different samples, but irrespective of the differences in 3D morphologies, the Bnl:GFP distribution consistently adopted the recipient ASP-specific contours (**Figure 1—figure supplement 2D–E"**). Collectively, these results showed that formation of the Bnl gradient depends on the ASP growth, and that the shapes of the gradient can dynamically adapt to the changing morphologies of the growing tissues.

A trachea-specific Bnl:GFP gradient might reflect a receptor-bound signal distribution. To examine this possibility, we created animals harboring a *btl:cherry^endo* knock-in allele using genome-editing technology (*Figure 2A*; *Figure 2—figure supplement 1A–C*; Materials and methods). As expected, Btl:Cherry expressed at physiological levels marked all tracheal cells, but formed a concentration gradient along the D-P axis of the ASP (*Figure 2B*; *Figure 2—figure supplement 1D*). This observation is consistent with an earlier report describing activation of *btl* transcription by Bnl signaling (*Ohshiro et al., 2002*). In trans-heterozygous *btl:cherry^endo*/*bnl:gfp^endo* animals, all the Bnl: GFP puncta in the ASP colocalized with Btl:Cherry and the receptor gradient coincided with that of the ligand gradient (*Figure 2C–D*; *Figure 2—figure supplement 1E,E'*). Most of the colocalized receptor-ligand puncta were compartmentalized apically in early- (αRab5 IF; ~76% of total puncta/ ASP examined) and late- (αRab7 IF; ~90% of total puncta/ASP) endosomes, but not in recycling endosomes or lysosomes (*Figure 2E–E"'*; *Figure 2—figure supplement 2A–C*). These results suggested that the branch-specific Bnl:GFP gradient is receptor-bound, and mostly intracellular. Notably, a small number of receptor-ligand puncta were found at the basal side of ASP cells (arrows in *Figure 2C,E*). They were located mostly at the distal ASP tip cells and did not colocalize with any of the endosomal markers we examined (*Figure 2E*). These puncta on the exposed basal side of the ASP might represent Btl:Cherry-bound Bnl:GFP molecules prior to their receptor-mediated endocytosis.

## Cytonemes mediate target-specific dispersion of Bnl:GFP

Bnl molecules were expected to move through the extracellular space from the wing disc source to the ASP. To clearly visualize the extracellular distribution pattern of post-secretory Bnl:GFP prior to their endocytosis in the ASP, we employed a standard detergent-free immunofluorescence (EIF) protocol (see Materials and methods). This protocol was specifically developed for detecting extracellular dispersion profiles of morphogens (*Schwank et al., 2011*; *Strigini and Cohen, 2000*). An αGFP EIF assay identified a distinct steady-state surface-bound extracellular/externalized pool of Bnl:GFP molecules (*Figure 3A*; henceforth referred as Bnl:GFP^ex). Interestingly, even though we used a highly sensitive Gallium arsenide phosphide detector for imaging (see Materials and methods), endogenous levels of EIF-stained Bnl:GFP^ex puncta appeared to emit very weak GFP-fluorescence, which was rapidly quenched during high magnification (40X) imaging. However, this property of GFP-tagged molecules enabled us to estimate the levels of externalized/extracellular Bnl:GFP (Bnl:GFP^ex), distinguishing them from the brightly fluorescent but non-EIF stained intracellular Bnl:GFP puncta in the ASP (henceforth referred as Bnl:GFP^in) (*Figure 3A*).

Most importantly, this experiment showed that although the EIF-stained Bnl:GFP^ex molecules represented secreted signals in the extracellular space, they did not disperse randomly in all directions. Instead, they were asymmetrically localized and only detected on the surfaces of signal producing cells and of the recipient ASP (*Figure 3A,A'*). Bnl:GFP^ex were highly enriched on the ASP and appeared to form a gradient on the recipient surface, in exactly the same way as the endocytosed Bnl:GFP^in molecules (*Figure 3A"*). In contrast to the asymmetric Bnl:GFP^ex distribution, a secreted-GFP construct containing a sfGFP tag and an N-terminal signal peptide derived from the Bnl sequence (secGFP; see Materials and methods) randomly dispersed in the extracellular space surrounding the disc *bnl*-source (*Figure 3B–B"*; *Figure 3—figure supplement 1A*). These results showed that Bnl:GFP moved target-specifically in polarized fashion.

The polarized distribution pattern of Bnl:GFP^ex was consistent with a potential cytoneme-mediated direct transport of the molecules. Previous studies showed that ASP cells project receptor-containing cytonemes to establish direct membrane contacts with the wing disc Bnl source. These cytonemes were shown to be essential for induction of pMAPK signaling in the ASP (*Roy et al., 2014*). However, Bnl transport through cytonemes was never visualized before. To examine if Bnl: GFP^ex molecules localized on the surface of the ASP cytonemes, we carried out EIF assays with either αGFP or αBnl antibody on CD8:Cherry-marked ASPs of *bnl:gfp^endo* homozygous larvae. These assays showed that the EIF-stained Bnl:GFP^ex puncta were highly enriched on cytonemes emanating from the distal ASP tip (N > 50 ASP, *Figure 3C–E*; *Figure 3—figure supplement 1B–B"*).

However, as observed earlier, many of these puncta were poorly fluorescent under standard confocal fluorescence imaging conditions. Therefore, to clearly visualize the endogenous Bnl:GFP molecules on cytonemes, we employed an enhanced gain super-resolution detection method (see Materials and methods). This imaging method revealed that the surfaces of the ASP and ASP

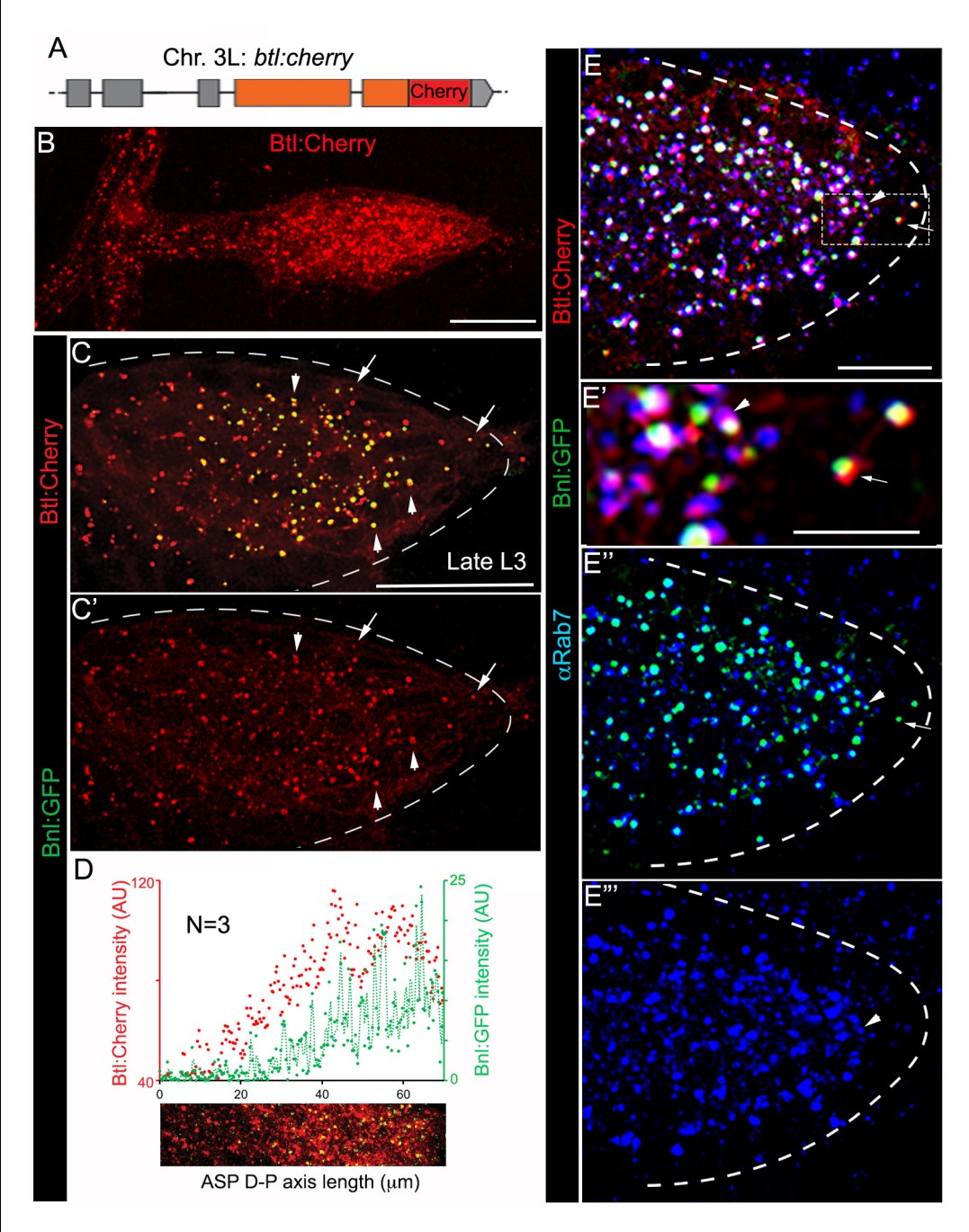

**Figure 2.** A receptor-bound Bnl:GFP gradient. (**A**) A schematic map of *btl:cherry^endo* knock-in allele; orange box, coding exon; grey box, non-coding exon; line, introns. (**B**) Btl:Cherry, expressed at physiological levels from the *btl:cherry^endo* allele, was detected as densely packed puncta marking the ASP cell membrane. (**C,C'**) Bright colocalized Bnl:GFP-Btl:Cherry puncta visible at low exposure imaging, 100% of the Bnl:GFP puncta in the ASP are Btl:Cherry bound; (**C'**) only red channel shown. (**D**) Graded Btl:Cherry expression and distribution in the ASP coincided with the Bnl:GFP gradient in the ASP; N = 3 independent samples; lower panel, an example of the region of ASP used for generating the intensity plot in the upper panel; AU, arbitrary unit. (**E-E'''**) About 90 ± 1.6% colocalized Bnl:GFP-Btl:Cherry puncta in the ASP also localized in αRab7-marked endosomes (arrowhead); (**E**) merged colors; (**E'**) zoomed in region marked by dashed box in E; (**E''**) only green and blue channels showing endosome localized Bnl:GFP in the ASP; (**E'**) only blue channel. (**C,C',E-E'''**) arrow, basally localized Bnl:GFP-Btl: Cherry puncta that were not in the endosomes; arrowhead, apically localized receptor-ligand puncta that were also in the endosomes; all images, Z-projection of 5–6 selected optical planes; dashed line, ASP outline. (C–

*Figure 2 continued on next page*

*Figure 2 continued*

E''') *bnl:gfp^{endo}/btl:cherry^{endo}* larvae. (B–E''') fixed samples. (B,D) Z-projection of 30–40 optical planes. Scale bars, 30 µm; 10 µm (E); 5 µm (E').

DOI: https://doi.org/10.7554/eLife.38137.012

The following source data and figure supplements are available for figure 2:

**Source data 1.** Data for the intensity profile plots of Bnl:GFP and Btl:Cherry in *Figure 2D*.
DOI: https://doi.org/10.7554/eLife.38137.015

**Source data 2.** Data for the intensity profile plot of Btl:Cherry in *Figure 2—figure supplement 1D*.
DOI: https://doi.org/10.7554/eLife.38137.016

**Source data 3.** Numerical data showing frequency of colocalization of Bnl:GFP puncta in the ASP with Btl:Cherry, early-, late-, and recycling- endosomes, and lysosomes in *Figure 2C,E*, and *Figure 2—figure supplement 2A,B,C*.
DOI: https://doi.org/10.7554/eLife.38137.017

**Figure supplement 1.** Generating genome-edited flies harboring a *btl:cherry^{endo}* allele.
DOI: https://doi.org/10.7554/eLife.38137.013

**Figure supplement 2.** Intracellular distribution of the receptor-bound Bnl:GFP puncta.
DOI: https://doi.org/10.7554/eLife.38137.014

cytonemes were densely populated with a high concentration of Bnl:GFP molecules, which appeared as small sub-resolution-sized puncta of <200 nm in diameter (henceforth referred as nanopuncta due to their sub-resolution size; arrowheads in *Figure 3F–F"*, *Figure 3—figure supplement 2A–A"'*). These nanopuncta were also enriched on the source cells, especially around the bouton-like sites of contact between the cytonemes and the source (*Figure 3F*). An αGFP EIF staining assay in combination with super-resolution imaging showed that these Bnl:GFP nanopuncta included both surface-bound molecules (EIF-stained) and intracellular molecules (non-EIF stained) (*Figure 3—figure supplement 2B,B'*). Occasionally, several nanopuncta aggregated to form bright punctal clusters on the cytonemes, the fluorescence of which was likely to be easily detectable under our standard confocal fluorescence imaging conditions (arrows in *Figure 3F',F"*). Although the molecular and cellular nature of the nanopuncta and their clusters on the cytonemes are unclear, these results provided substantial evidence for a selective distribution of Bnl:GFP molecules along the surface of cytonemes.

Native, untagged Bnl could also be detected on cytoneme surfaces using the αBnl antibody. The αBnl EIF assay showed that a high concentration of externalized Bnl (Bnl^{ex}) molecules selectively decorate actin-rich ASP cytonemes and the ASP surface of *w⁻* wild-type larvae (*Figure 3G*). The distribution of Bnl^{ex} was asymmetrical, exactly as observed for Bnl:GFP on CD8:Cherry-marked ASPs (*Figure 3C*). These results suggested that Bnl molecules, irrespective of the presence or absence of fluorescent tags, specifically localize on the surface of the ASP cytonemes for inter-organ transport. Since efficient inter-tissue movement of Bnl molecules along the surface of the recipient cytonemes might require receptor binding, we imaged ASP cytonemes in *btl:cherry^{endo}* larvae. Densely packed Btl:Cherry molecules were visualized on the surface of the ASP cytonemes (*Figure 3—figure supplement 3A*). An αBnl-EIF assay in *btl:cherry^{endo}* larvae showed that the native Bnl^{ex} puncta colocalized with the Btl:Cherry puncta on the cytonemes and on ASP surfaces (*Figure 3H*). Receptor-bound Bnl:GFP puncta were also observed on cytonemes during live imaging (*Figure 3—figure supplement 3B–C'*; N = 30). Thus, Bnl is transported by cytonemes from the source to the ASP in a receptor-bound state.

Cytonemes exchange signals by establishing direct physical contacts between the source and recipient cells (*Chen et al., 2017*; *González-Méndez et al., 2017*; *Huang and Kornberg, 2015*; *Roy et al., 2014*). These signaling contacts between the ASP and the *bnl* source were originally discovered using a membrane GFP-reconstitution method (GFP reconstitution across synaptic partners, GRASP) (*Huang and Kornberg, 2015*; *Roy et al., 2014*). As shown in *Video 3*, an improved GRASP technique, 'Synaptobrevin-GFP-reconstitution-across-synaptic-partners (*syb*-GRASP)' revealed multiple functional contact sites between each of the ASP cytonemes and the *bnl*-source. To estimate the contact-dependency of the cytoneme-mediated Bnl reception, we combined the *syb*-GRASP experiment with the αBnl EIF assay. This experiment showed that the actin-rich (phalloidin-stained) ASP cytonemes that contacted the disc *bnl*-source received a significantly higher number of Bnl^{ex} puncta in comparison to the non-contacting cytonemes (*Figure 3I,I'*). These observations, together with the

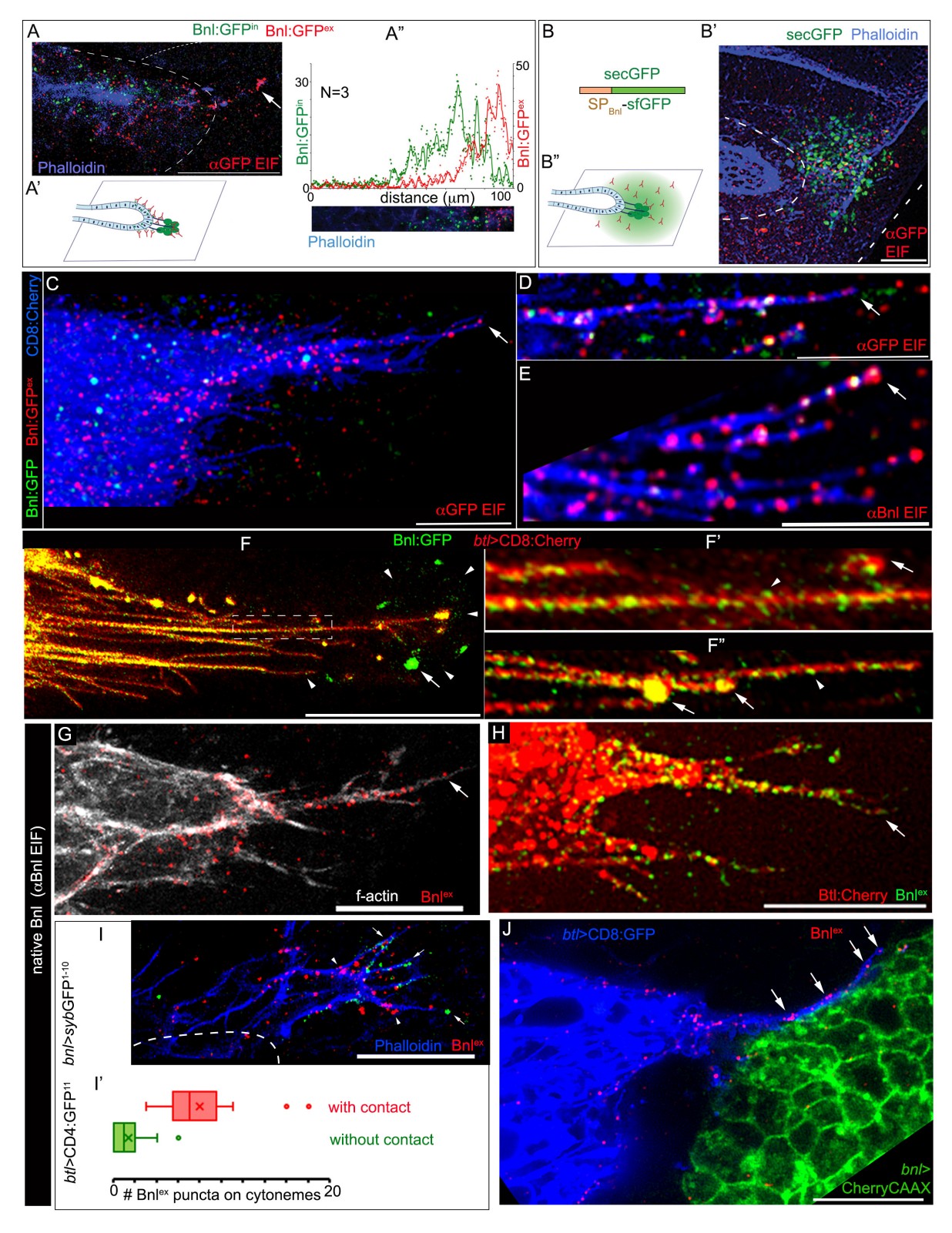

**Figure 3.** ASP cells receive Bnl:GFP through Btl-containing cytonemes. (**A-A''**) An αGFP-based EIF assay under the detergent-free conditions (Materials and methods), identified externalized Bnl:GFP (red, Bnl:GFPex) that localized specifically on the surfaces of the ASP, actin-rich ASP protrusion (arrow), and, at a low level, on wing disc signal producing cells (area within the dotted line); (**A'**) concentration gradient of surface-localized Bnl:GFPex similar to that of the intracellular Bnl:GFPin along the D-P axis of the ASP (N = 3 independent samples); bright Bnl:GFPin puncta were unrecognized by the EIF

*Figure 3 continued on next page*

*Figure 3 continued*

method; EIF-stained Bnl:GFP[ex] puncta (red) emitted poor GFP fluorescence. (**B-B''**) Control experiment for (**A**), showing random non-specific distribution (red, αGFP EIF) of extracellular secreted-GFP (secGFP; drawing in B; Materials and methods) when expressed from the disc *bnl*-source (green; see *Figure 3—figure supplement 1A*); (**A',B''**) drawings explaining distribution patterns in A,B' respectively; (**A,B'**) blue, phalloidin-Alexa-647 staining of f-actin marked cell outlines. (**C–E**) Bnl:GFP[ex] puncta (red; arrow) localized on the surface of CD8:Cherry-marked cytonemes (blue) emanated from the ASP (*btl-Gal4, UAS-CD8:Cherry/+; bnl:gfp[endo]*); (**C,D**) αGFP-EIF; (**E**) αBnl-EIF. (**F-F'**) Enhanced-gain super-resolution images (Materials and methods), showing a sub-resolution distribution of faint Bnl:GFP molecules (nanopuncta) in live CD8:Cherry-marked ASP and ASP cytonemes (see *Figure 3—figure supplement 2A–B'*); genotype: *btl-Gal4, UAS-CD8:Cherry/+; bnl:gfp[endo]*; (**F'**) zoomed-in area indicated by the ROI box in (**F**); (**F''**) cytonemes from a different sample showing difference of nano-puncta (arrowheads) and large puncta representing clusters (arrow) of nano-puncta. (**G**) An αBnl EIF showing biased distribution of the native externalized Bnl puncta (Bnl[ex], arrow) in wild type (*w[-]*) background; grey, phalloidin-Alexa-647. (**H**) Bnl[ex] puncta colocalized with endogenous Btl:Cherry on cytonemes extended from the *btl:cherry[endo]* ASP. (**I,I'**) Comparison of Bnl[ex] (αBnl EIF, red; arrowhead) localization on the actin-rich ASP cytonemes (blue, phalloidin-Alexa 647) with and without a direct contact (green dots, arrow) with the disc *bnl*-source; genotype: *btl-Gal4/lexO-nsyb:GFP[1-10], UAS-CD4:GFP[11]; bnl-LexA/+*; (**I'**) a graph showing the comparative numbers; N = 14 ASP; p<0.0001 (two-tailed t-test). (**J**) A high-resolution image showing contacts between the ASP cytonemes (blue) and wing disc source (green) and enrichment of native externalized Bnl (red; Bnl[ex], detected with αBnl EIF) at these contact points; genotype: *btl-Gal4, UAS-CD8:GFP/+; bnl-LexA, lexO-CherryCAAX/ +*. Scale bars, 20 μm.

DOI: https://doi.org/10.7554/eLife.38137.018

The following source data and figure supplements are available for figure 3:

**Source data 1.** Numerical data for measuring and plotting intensity profiles of Bnl:GFP[ex] (αGFP EIF) in *Figure 3A''*.
DOI: https://doi.org/10.7554/eLife.38137.022

**Source data 2.** Numerical data for comparing the numbers of Bnl[ex] puncta on cytonemes with and without a direct contact with the disc *bnl* source in *Figure 3I'*.
DOI: https://doi.org/10.7554/eLife.38137.023

**Figure supplement 1.** Cytoneme-mediated Bnl:GFP transport.
DOI: https://doi.org/10.7554/eLife.38137.019

**Figure supplement 2.** Distribution of Bnl:GFP on cytonemes and cell surfaces.
DOI: https://doi.org/10.7554/eLife.38137.020

**Figure supplement 3.** Live imaging of Bnl:GFP and Btl:Cherry on cytonemes.
DOI: https://doi.org/10.7554/eLife.38137.021

previous reports on the effect of loss of cytoneme contacts on pMAPK signaling in the ASP (*Roy et al., 2014*), provided substantial evidence that cytoneme-mediated Bnl exchange is contact-dependent.

Molecular reconstitution of split GFP between ASP cytonemes and source cells suggested that the signaling contacts between the cytoneme and source cell membranes juxtapose within a 20 nm gap distance (*Kornberg and Roy, 2014*). To directly visualize and resolve Bnl exchange at these cytoneme contacts, an αBnl EIF assay was combined with super-resolution imaging (see Materials and methods) of the tissues that had both the source and recipient ASP cells marked with distinct membrane-tagged proteins (*Figure 3J*). The images revealed that ASP cytonemes traverse along the surface of the source cells, with each cytoneme establishing contacts with multiple signal-producing cells. Although the producing cells occupy a large area, the externalized Bnl proteins were specifically enriched at the cytoneme contact sites. This observation suggests that the producing cells selectively release Bnl at these contact sites. These ASP cytonemes also localized Bnl[ex] puncta

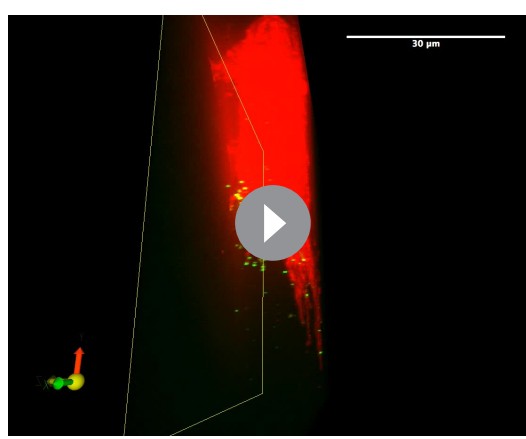

**Video 3.** 3D rendered views of the CD8:Cherry-marked ASP showing that each of the ASP cytonemes established multiple physical contacts (green dots) with the underlying wing disc *bnl* source (unmarked). Cytoneme contact sites were marked by *syb*GRASP, a GFP-reconstitution method, where the ASP expressed CD4:GFP[11] and the wing disc *bnl*-source expressed *syb:GFP[1-10]*; genotype: *btl-Gal4, UAS-CD8:Cherry/ LexO-syb:GFP[1-10], UAS-CD4:GFP[11]; bnl-LexA/+*.
DOI: https://doi.org/10.7554/eLife.38137.024

on their shafts, suggesting that the signal might be transported from the contact sites to the ASP along the surface of the cytonemes. A similar retrograde movement of Dpp was previously observed through ASP cytonemes (*Roy et al., 2014*).

## Cytoneme-mediated transport is essential for generation of the Bnl gradient

To examine whether cytoneme-mediated reception of Bnl:GFP is essential for generating the long-range Bnl:GFP gradient in the ASP, we generated mosaics of marked cytoneme-deficient mutant clones within the ASP of *bnl:gfp^endo* larvae by knocking down several known cytoneme-regulator genes such as *diaphanous* (*dia*), a formin homolog, *neuroglian* (*nrg*), an L1-cell adhesion molecule (*Roy et al., 2014*), and *singed* (*sn*), a Fascin homolog and a regulator of tracheal filopodia (*Okenve-Ramos and Llimargas, 2014*). To assess Bnl reception in the mutant clones, we compared Bnl:GFP levels within the mutant area (inside of the clone) with that in the wild-type (WT) neighboring areas at an equivalent position (outside of the clone) in the same ASP. The ratios of Bnl:GFP concentration outside to inside of the marked wild-type (WT) control clones were ~1, suggesting no significant differences in the signal uptake within the marked (inside) and unmarked (outside of the clone) WT areas (*Figure 4A,E*). However, knockdown of cytoneme-regulator genes in clones of cells significantly decreased Bnl:GFP levels in the mutant cells, but not in the neighboring WT cells (*Figure 4B–E*; *Figure 4—figure supplement 1A*). Consequently, the ratio of Bnl:GFP concentration outside (WT area) to inside (cytoneme deficient) of the mutant clones increased significantly compared to the WT control condition (*Figure 4E*). These results showed that cytonemes are essential for the ASP cells to receive Bnl:GFP from the wing disc source.

The distribution profile of Bnl:GFP in *btl*-LOF clones further suggested that cytonemes are the only major routes for Bnl gradient formation. As illustrated in *Figure 4F,F'*, a cytoneme-mediated mechanism of dispersion, which is receptor-dependent, would create cell-autonomous loss of receptor-bound Bnl:GFP^ex on the surface of a *btl*-LOF clone. This predicted distribution pattern is different from that of a passive extracellular Bnl:GFP diffusion, which would be expected to result in unperturbed levels of the extracellular EIF-stained Bnl:GFP^ex on the receptor-LOF clone. Indeed, *RNAi*-mediated knockdown of *btl* in clones of ASP cells led to a significant reduction of Bnl:GFP^ex on the mutant clonal areas compared to the WT neighboring cells (*Figure 4G,G'*). Clearly, the distribution pattern of Bnl:GFP^ex was consistent with a cytoneme-dependent mechanism of gradient formation. Note that an autonomous reduction in intracellular Bnl:GFP (non-EIF stained) was also observed in the mutant clones, but such reduction in intracellular Bnl:GFP levels was expected irrespective of the mechanism of signal dispersion, and did not affect our conclusions. Furthermore, in the ASP stalk, where normal levels of Btl are low, generation of *btl* overexpressing (*btl*-GOF, gain of function) clones induced extension of long polarized cytonemes that projected toward the underlying disc *bnl*-source. These cytonemes were enriched with EIF-stained Bnl:GFP^ex on their surfaces (*Figure 4H,H'*) and the mutant clones had a higher concentration of Bnl:GFP^ex than their neighbors (*Figure 4—figure supplement 1B*). This result provided direct evidence that the induction of cytonemes in ASP cells increased cell-autonomous contact-dependent Bnl:GFP uptake.

## Signaling through a graded number of cytonemes generates the Bnl:GFP gradient in the ASP

How might cytoneme-mediated local signaling in each ASP cell generate a long-range signal gradient across the entire ASP epithelium? One possibility is that tracheal cells, relative to their distances from the *bnl*-source, produce different numbers of Bnl-specific cytonemes and/or establish variable numbers of signaling contacts with the disc *bnl*-source. To examine this possibility, we live imaged randomly generated CD8:GFP-marked clones in different positions of the ASP and analyzed the orientation and number of cytonemes with different lengths they extend (*Figure 5A–A"*). In addition, to compare the frequency of signaling contacts among cytonemes emanating from different regions of the ASP, we devised a mosaic analysis method for generation of clonal *syb*-GRASP. We expressed *syb*-GFP^1-10 in the *bnl*-source and induced small (~1–4 cells) randomly localized CD8:RFP-marked clones across the ASP epithelium that also expressed CD4:GFP^11 (*Figure 5D,D'*; see Materials and methods). In this technique, when cytonemes emanating from the clones in different parts of the ASP established contact with the disc *bnl*-source, GFP was reconstituted at their contact sites. For

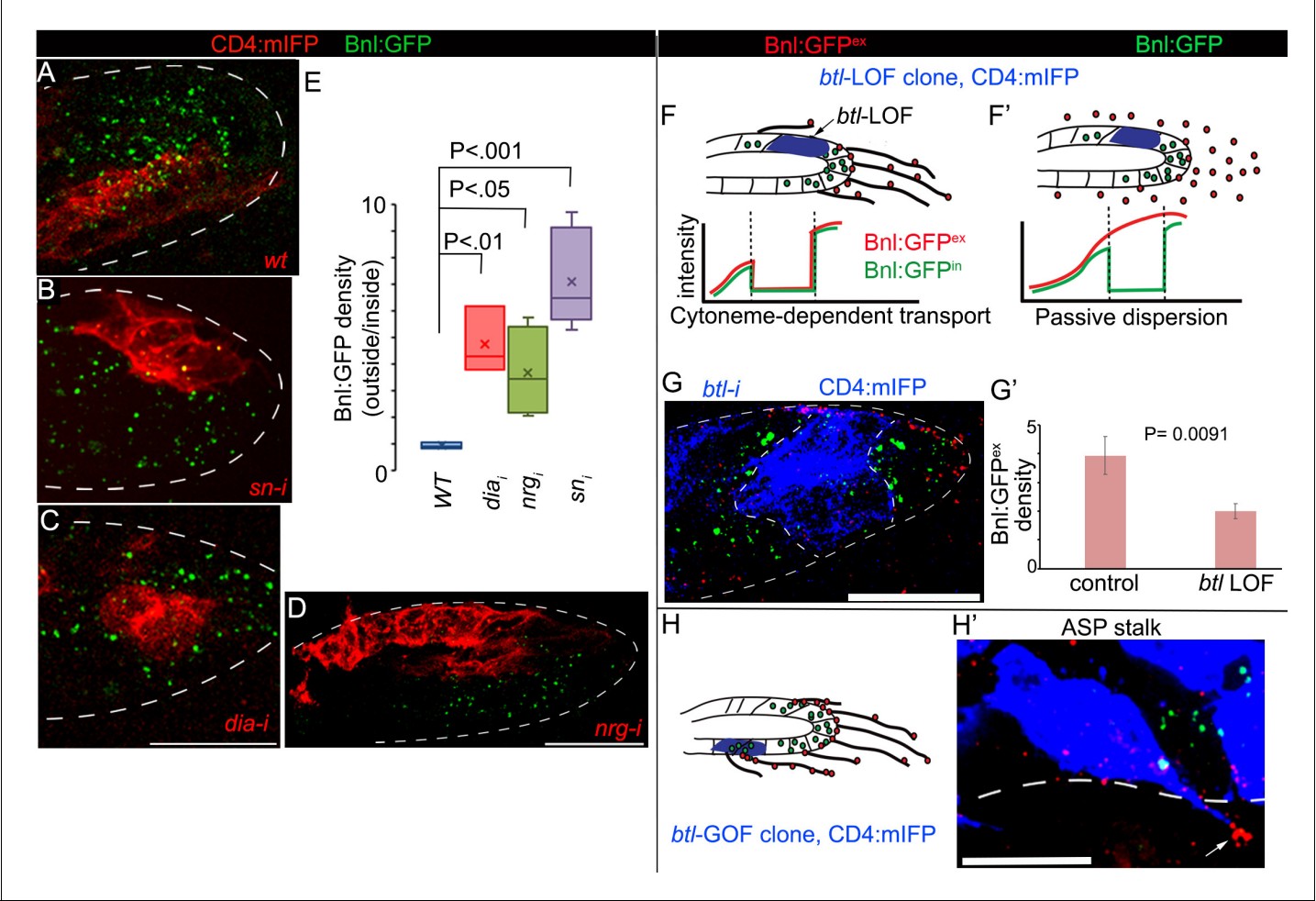

**Figure 4.** Cytonemes are essential for tissue-specific Bnl:GFP gradient profile. (**A-D**) Representative images of Bnl:GFP uptake in CD4:mIFP-marked (red) clones in the ASP: (**A**) *wt;* (**B**) *snRNAi,* (**C**) *diaRNAi;* (**D**) *nrgRNAi.* (**E**) A box plot comparing the ratios of Bnl:GFP concentration outside (unmarked WT area) to inside (marked mutant area) of the CD4:mIFP-marked cytoneme-deficient mutant clones, expressing either *diaRNAi* (N = 3), *nrgRNAi* (N = 4), or *snRNAi* (N = 6) with that of the control marked *WT* clones (N = 3); P values, two-tailed t-test. (**F,F'**) Schematic models predicting differences expected in distribution patterns of EIF-stained Bnl:GFP$^{ex}$ (red) on *btl* LOF clones (blue) either in diffusion-based (**F'**) or cytoneme-dependent (**F**) Bnl:GFP transport mechanism. Intracellular Bnl:GFP (green, Bnl:GFP$^{in}$) level is expected to be reduced in both modes of signal transport. (**G**) Representative image showing autonomous loss of distribution of Bnl:GFP$^{ex}$ (red, αGFP EIF-stained) in CD4:mIFP-marked (blue) *btl* LOF (*btl*RNAi) clone. (**G'**) Plot comparing relative Bnl:GFP$^{ex}$ (red) density inside (mutant area) versus outside (WT area, control) of *btl* LOF clones (N = 3 independent ASPs); P values, two-tailed t-test. (**H,H'**) Representative image showing enrichment of αGFP EIF stained Bnl:GFP$^{ex}$ (red) on long cytonemes (arrow) projected from *btlGOF* clones in the ASP stalk. Genotypes: *hs-FLP/+; btl>y$^+$>Gal4/+; UAS-CD4:mIFP, bnl:gfp$^{endo}$/+* (**A**) or *UAS-'x'-RNAi* (**B–G**) or *UAS-btl* (**H'**). Scale bars, 30 μm.

DOI: https://doi.org/10.7554/eLife.38137.025

The following source data and figure supplement are available for figure 4:

**Source data 1.** Numerical data for measuring the ratio of Bnl:GFP concentration outside:inside of the marked clones in *Figure 4E*.
DOI: https://doi.org/10.7554/eLife.38137.027
**Source data 2.** Numerical data for measuring Bnl:GFP$^{ex}$ levels between *btl* LOF clones and the neighboring *wt* area in *Figure 4—figure supplement 1G'*.
DOI: https://doi.org/10.7554/eLife.38137.028
**Source data 3.** Numerical data for measuring the relative Bnl:GFP$^{ex}$ density in *btl* GOF clones and the neighboring *wt* area in *Figure 4—figure supplement 1B*.
DOI: https://doi.org/10.7554/eLife.38137.029
**Figure supplement 1.** Cytonemes are essential for Bnl:GFP gradient.
DOI: https://doi.org/10.7554/eLife.38137.026

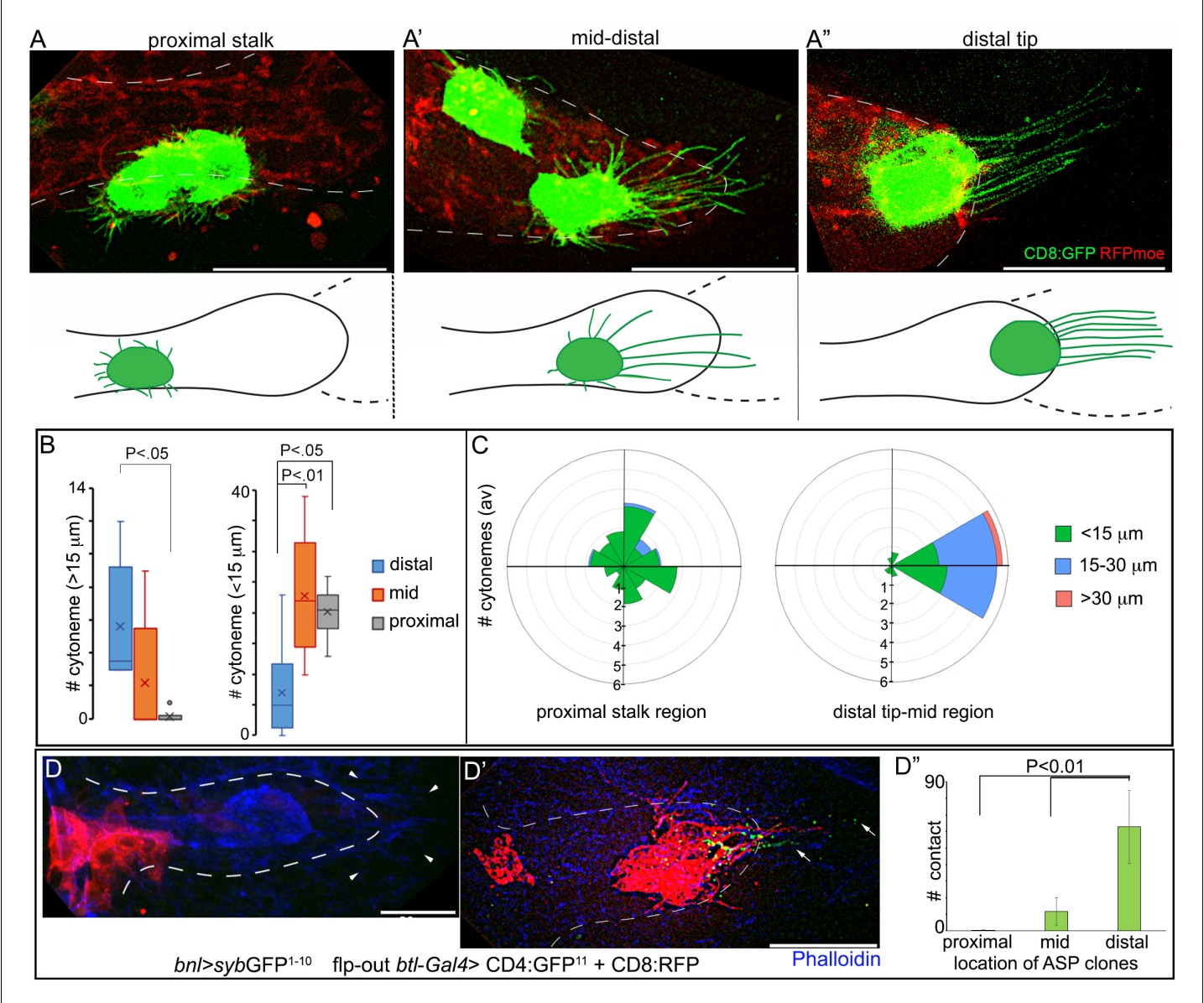

**Figure 5.** A graded pattern of cytoneme formation in the ASP. (A-C) Live images of ASPs showing variability in the number of long, oriented cytonemes emanating from the CD8:GFP-marked clones located in different ASP zones: (A) proximal stalk (>7th cells from distal tip of 12 cell long D-P axis); (A') mid-region (4th-7th cells along the D-P axis); (A'') distal tip (1st-3rd cells along the D-P axis); red, *btl*>RFP_moe; lower panels, drawing showing approximate location of the clones and the corresponding cytoneme patterns, dashed line, *bnl*-source; (B) graphs comparing the number of long (>15 μm) and short (<15 μm) cytonemes from clones in the distal tip (N = 8), mid (N = 5), and proximal (N = 6) regions in the ASP; (C) rose plots comparing number and orientation of cytonemes of different lengths emanated from CD8:GFP-marked clones at distal tip-to-mid region (1st-7th cell along D-P; N = 7) and proximal stalk (>7 cells away from distal tip along D-P axis; N = 11) of the ASP; genotype: *hs-FLP/+; UAS-mCD8:GFP/+; btl>y⁺>Gal4, btl-mRFP1moe/+*. (D-D'') Clones of cells expressing CD8:RFP and CD4:GFP[11] from tip-, mid-, and stalk- region of the ASP established variable number of contacts with *syb*GFP[1-10] expressing disc *bnl*-source; green puncta, GFP-reconstitution signal representing contact sites between the ASP cells and the source; (D'') plots comparing the number of contacts established by cells from different regions of the ASP with the disc *bnl* source; blue, phalloidin-Alexa 647; genotype: *hs-FLP/+; btl>y⁺>Gal4/lexO-nsyb:GFP[1-10], UAS-CD4:GFP[11]; bnl-LexA/UAS-CD8:RFP*. All panels except D-D'', live imaging. Scale bars, 30 μm.

DOI: https://doi.org/10.7554/eLife.38137.030

The following source data and figure supplement are available for figure 5:

**Source data 1.** Numerical data for the number of long (>15 μm) and short (<15 μm) cytonemes from *wt* clones in different regions of the ASP in *Figure 5B*.

DOI: https://doi.org/10.7554/eLife.38137.032

**Source data 2.** Numerical data for the number of cytonemes oriented in different directions from *wt* clones in *Figure 5C*.

*Figure 5 continued on next page*

*Figure 5 continued*

DOI: https://doi.org/10.7554/eLife.38137.033

**Source data 3.** Numerical data for comparing the number of contacts established by cytonemes emanated from clones at different regions of the ASP in *Figure 5D"*.

DOI: https://doi.org/10.7554/eLife.38137.034

**Figure supplement 1.** A graded pattern of cytoneme contacts.

DOI: https://doi.org/10.7554/eLife.38137.031

assessing the steady-state number of contacts established by different parts of the ASP with the signal source, ASPs containing only 1–2 small clones were considered.

These experiments showed that the stalk region of a third instar larval ASP, beyond 7 cell-distances along the D-P axis (~12 cells long) from the ASP tip (*Figure 1A"*), produced only randomly oriented short cytonemes. These cytonemes rarely contacted the *bnl*-source (*Figures 5A,B,C,D–D"*). This region also received lower levels of Bnl:GFP (*Figure 1A",E*). The number of long polarized cytonemes that oriented toward the *bnl*-source and established direct contacts with the source increased gradually in the mid (4th-7th cells from the ASP tip along the D-P axis) and the distal tip cells (1–3 cells from the tip along the D-P axis) (*Figure 5A',A",B,C,D',D"*; *Figure 5—figure supplement 1A–A"*). Thus, long oriented cytonemes that could establish contacts with the Bnl source were produced in a graded fashion across the length of the ASP. A similar gradient in the number of signaling contacts was also observed from cytonemes produced across the vertical axis of the tubular tissue. In a *syb*-GRASP experiment between the disc and ASP, ASP cells in the lower layer that is proximal to the *bnl*-source had a higher frequency of cytoneme contacts compared to the upper cell layer (*Figure 5—figure supplement 1B*; *Videos 3*, *4* and *5*). These results suggested that differential levels of signal uptake through a graded number of

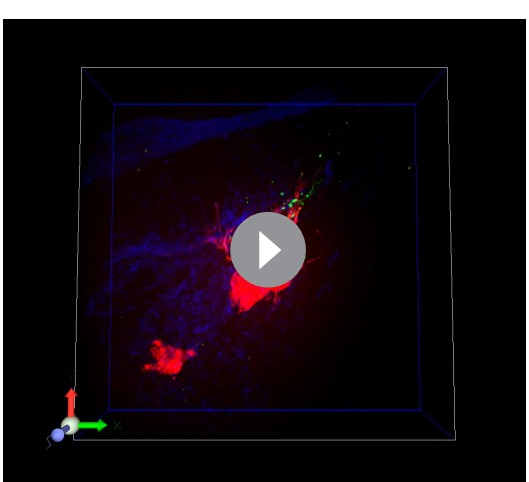

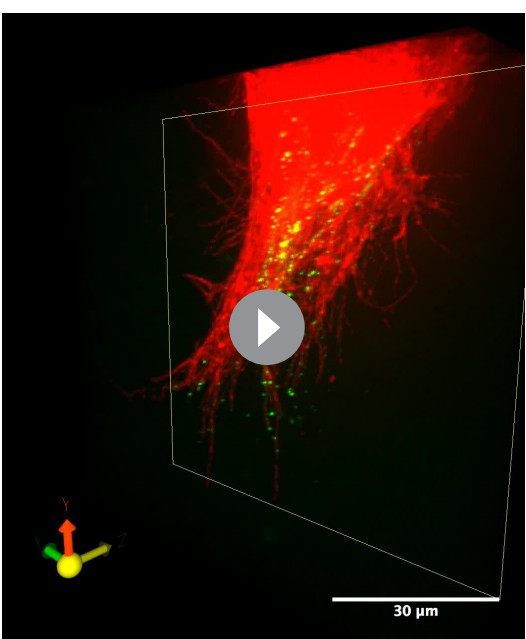

**Video 4.** 3D rendered views of the ASP and wing disc showing that the CD8:RFP- marked clones of cells in the ASP tip had higher number of cytoneme-mediated contacts (green) with the *bnl*-source than the clones in the ASP stalk; ASP outline was marked by phalloidin (blue); CD8:RFP marked clones of ASP cells at different positions expressed CD4:GFP[11] and disc *bnl*-source (unmarked) expressed *sybGFP[1-10]*. GFP-reconstitution indicated physical contacts between the wing disc source and the clonal part of the ASP; genotype: *hs-FLP/+; btl>y⁺>Gal4/lexO-nsyb:GFP[1-10], UAS-CD4:GFP[11]; bnl-LexA/UAS-CD8:RFP*.

DOI: https://doi.org/10.7554/eLife.38137.035

**Video 5.** 3D rendered views of the CD8:Cherry-marked ASP showing the differences in the number of cytoneme contacts with the disc *bnl*-source from the upper and lower layers of ASP cells; cytonemes from the lower layer ASP cells, which are proximal to the underlying disc source (unmarked) have higher number of contacts than the upper layer ASP cells; cytoneme contact sites (green) were marked by *syb*GRASP; genotype: *btl-Gal4, UAS-CD8:Cherry/LexO-syb:GFP[1-10], UAS-CD4:GFP[11]; bnl-LexA/+*.

DOI: https://doi.org/10.7554/eLife.38137.036

cytonemes give rise to the tissue-specific shapes of the Bnl:GFP gradient.

## Bnl induces concentration-dependent differential gene expression in the ASP

How does a steady state pattern of cytonemes across the ASP epithelium develop? The gradient of numbers of Bnl-receiving cytonemes correlated with the spatial gradient of Bnl (*Figures 5A–B;1E,I; 2D*). Thus, the zones of variable numbers of ASP cytonemes might reflect differential target gene activities in response to the Bnl gradient. Therefore, we examined the correlation of the Bnl:GFP gradient with the spatial expression domains of several known Bnl target genes, including *pntP1* and *sprouty* (*sty*) (*Hacohen et al., 1998*; *Ohshiro et al., 2002*). We also examined expression of the Ets family transcriptional repressor *yan*, which is suppressed by Ras/MAPK signaling (*Ohshiro et al., 2002*), and of *cut*, a homeodomain transcription repressor. Although *cut* is not known to be a target of Ras/MAPK signaling, this gene was shown to be expressed in the larval spiracular branch and in the ASP (*Ohshiro et al., 2002*; *Pitsouli and Perrimon, 2013*; *Rao et al., 2015*).

The D-P axis of a third instar larval ASP consists of ~12 cells. We found that ASP cells within a three-cell diameter domain from the distal tip along the D-P axis, which received the highest concentration of Bnl:GFP, expressed *sty* (*Figure 6A,E*; *Figure 6—figure supplement 1A*; *Figure 6—figure supplement 2A,B*). A broader zone of ~6–7 cell-diameters from the tip along the D-P axis received high-to-moderate levels of Bnl:GFP and induced *pntP1* (*Figure 6B,E*; *Figure 6—figure supplement 1A*; *Figure 6—figure supplement 2C, D*). In contrast, the proximal stalk of the ASP from the 7th to 12th cell away from the tip along the D-P axis received negligible levels of Bnl:GFP. In this ASP domain, *cut* and *yan* expression gradually increased with increasing distance from the source (*Figure 6C–E*; *Figure 6—figure supplement 1A*; *Figure 6—figure supplement 2E-H*). Thus, the gradients of *cut* and *yan* expression had the exact opposite pattern as the *pntP1* and Bnl:GFP gradients. This concentration-dependent activity was also observed across the Z-axis (*Figure 1A"*). ASP cells in the lower Z-sections that are proximal to the underlying *bnl*-source expressed higher levels of *sty* and *pntP1* relative to the upper layer cells situated away from the disc source (*Figure 6F*; *Figure 6—figure supplement 1B,C*). Thus, across the recipient ASP epithelium, high levels of Bnl:GFP correlated with *sty* expression, and gradually decreasing levels of Bnl:GFP further away from the source correlated with the expression zones of *pntP1*, *cut*, and *yan*, respectively (*Figure 6E,F*). Similar spatial patterning in the gene-expression zones of *sty* and different ETS family proteins is known to be induced by the vertebrate FGF8 morphogen in mouse neuroectoderm (*Toyoda et al., 2010*).

Previously Bnl was considered to be a chemoattractant (*Lebreton and Casanova, 2016*; *Ochoa-Espinosa and Affolter, 2012*), which can only elicit a binary signaling response. In contrast, our results suggested that Bnl activated multiple target genes at its various concentrations similar to a morphogen. To further examine whether the target genes were differentially expressed in response to different Bnl concentrations, we overexpressed cDNA-derived Bnl constructs either from the wing disc *bnl*-source or from small clones of cells within the recipient ASP. Remarkably, even while highly expressed from the wing disc source, Bnl:GFP moved only target-specifically from the disc source to the ASP (*Figure 6H,H'*). In comparison to the normal range of endogenous Bnl:GFP distribution and Bnl (pMAPK) signaling (*Figure 6G*), the spatial range of the overexpressed signal and the signaling expanded to all parts of the ASP, including the stalk (*Figure 6H*). The expanded range of signal led to retraction of the Cut and Yan expressing zone from within the stalk to only the farthest region in the TC, which received little or no Bnl:GFP from the disc source (*Figure 6H–I*). Moreover, ectopic clones of either Bnl:GFP or Bnl in the ASP stalk non-autonomously induced *sty* and *pntP1* and suppressed *cut* in the surrounding cells (*Figure 6J,K,L*; *Figure 6—figure supplement 1D*). These results showed that Bnl acts as a morphogen to induce concentration-dependent expression of different genes.

It was previously reported that PntP1 and Yan antagonize each other's expression (*Ohshiro et al., 2002*). To examine whether Cut and PntP1 also antagonize each other, we induced ectopic *pntP1* and *cut* mutant clones in the ASP. Reduction of PntP1 levels in the mutant *pntP1* LOF (*pntP1 RNAi*) clones in the ASP tip autonomously upregulated Cut expression within the clones, whereas overexpression of PntP1 in the ectopic *pntP1* GOF clones in the ASP stalk suppressed *cut* in the mutant cells (*Figure 6M,N*). On the other hand, ectopic *cut* GOF clones in the ASP tip suppressed *pntP1* and *cut* LOF clones (*cut RNAi*) in the proximal ASP stalk induced *pntP1* within the mutant cells (*Figure 6O,P*). Thus, PntP1 and Cut reciprocally inhibit each other's expression.

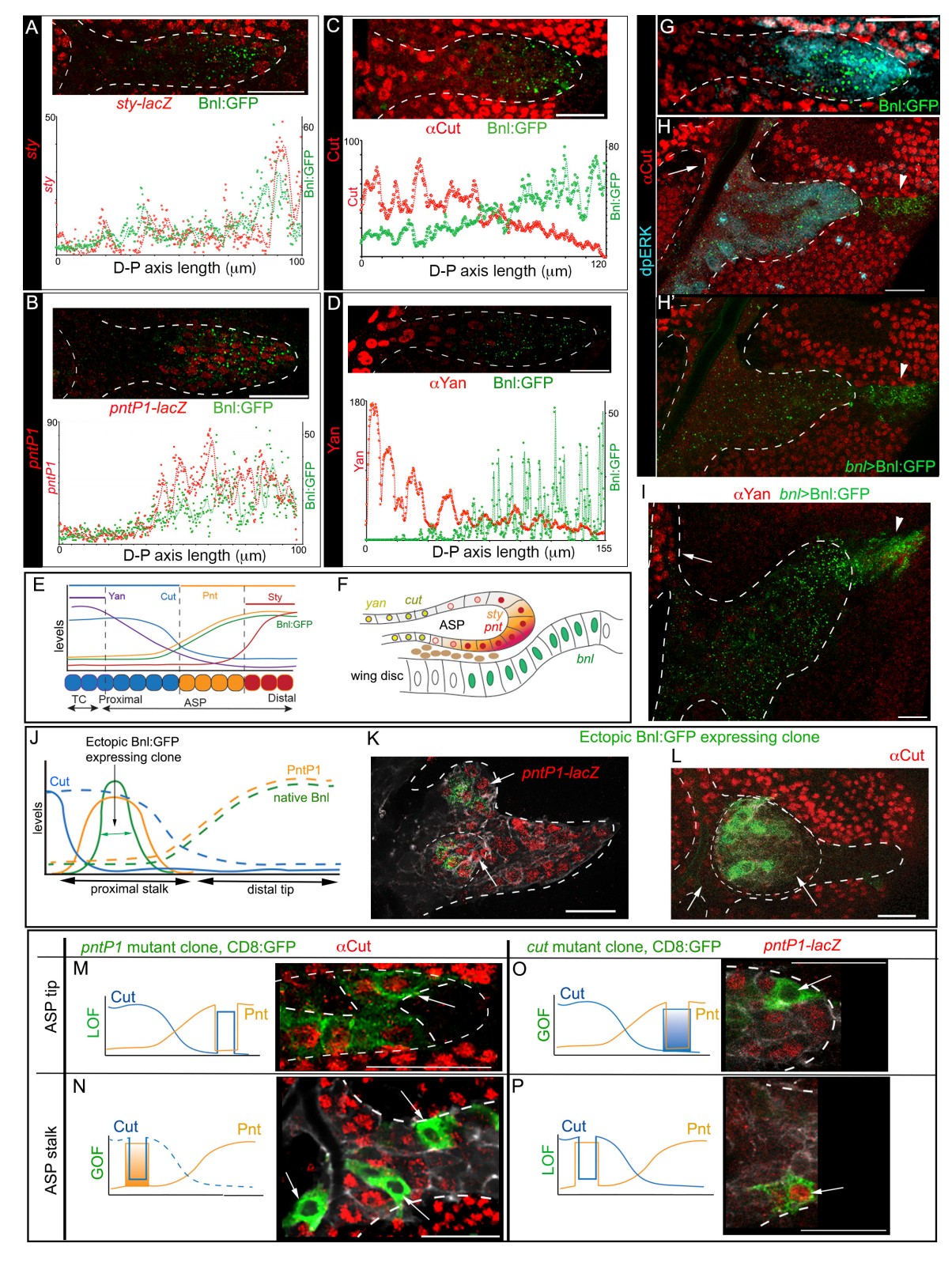

**Figure 6.** Bnl:GFP acts as a morphogen to activate concentration-dependent gene expression. (**A-D**) Representative images showing correlation of the Bnl:GFP gradient with spatial expression domains (red) of *sty* (A; *sty-lacZ*), *pntP1* (B; *pntP1-lacZ*), *cut* (C; αCut), and *yan* (D; αYan); lower panels, intensity plots along the D-P axis of the ASP. (**E**) Drawing depicting the expression domains of *sty*, *pntP1*, *cut*, and *yan* within 1st-3rd, 1st-7th, 7th-14th, and 11th-14th cells, respectively, from the distal tip of a 14 cell long D-P axis of ASP including TC (2 cell wide). (**F**) Drawing of a sagittal section of an ASP

*Figure 6 continued on next page*

*Figure 6 continued*

depicting relative gene expression domains in Z axis (see *Figure 6—figure supplement 1C*). (G) Correlation of spatial range of endogenous Bnl:GFP gradient with induction of pMAPK signaling (α-dpERK, blue) and Cut expression (α-Cut, red). (H–I) Overexpression of Bnl:GFP from the disc *bnl*-source (arrowhead) expanded the range of Bnl:GFP and pMAPK signaling (H,H') to the ASP stalk, suppressing/retracting the *cut* (H, arrow) and *yan* (I, arrow) domains to the farthest part that received little or no Bnl:GFP (arrow; red). (J) A hypothetical expression pattern (solid lines) of PntP1 (orange) and Cut (blue) induced by an ectopic Bnl:GFP (green) expressing clone in the ASP stalk; dashed lines, original native Bnl, PntP1, and Cut expression pattern. (K, L) Small Bnl:GFP overexpressing clones (green) at the ASP stalk non-autonomously suppressed *cut* (L; red; arrow), induced *pntP1* (K; red; arrow), and organized the surrounding cells to induce ectopic branch (dashed outlines); genetic crosses: *hs-FLP; btl>y+>Gal4, btl-mRFP1moe* X *UAS*-Bnl:GFP (L), or *UAS*-Bnl:GFP; *pntP1-lacZ/TM6* (K). (M–P) Cut and PntP1 reciprocally antagonized each other's expression (red); GOF/LOF, Gain/Loss of function; (M,N) *hs-FLP; UAS-mCD8GFP; btl>y+>Gal4,btl-mRFP1moe* X *UAS-pntRNAi* (M), or *UAS-PntP1* (N); (O) *hs-FLP; btl>y+>Gal4,btl-mRFP1moe* X *UAS-Cut, UAS-CD8:GFP; pntP1-lacZ/TM6*; (P) *hs-FLP; UAS-mCD8GFP; btl >y+>Gal4, btl-mRFP1moe* X *UAS-cutRNAi, pntP1-lacZ/TM6*. (M,O) ASP tip/mid-region; (N,P) ASP stalk/TC region; LOF, *RNAi*-mediated knockdown; arrows, the intended GOF/LOF clones. (A–P) white dashed line, ASP or clone outlines. (C,G,H, H',L,M,N) αCut; (A,B,K,O,P) anti-βGal for *pntP1-lacZ* (B,K,O,P) and *sty-lacZ* (A). Genotypes, see Materials and methods. Scale bars, 30 μm.

DOI: https://doi.org/10.7554/eLife.38137.037

The following source data and figure supplements are available for figure 6:

**Source data 1.** Data for the intensity profile plots of Bnl:GFP and *sty* (*sty-LacZ*) in *Figure 6A*.
DOI: https://doi.org/10.7554/eLife.38137.040
**Source data 2.** Data for the intensity profile plots of Bnl:GFP and *pntP1* (*pntP1-LacZ*) in *Figure 6B*.
DOI: https://doi.org/10.7554/eLife.38137.041
**Source data 3.** Data for the intensity profile plots of Bnl:GFP and *cut* (αCut) in *Figure 6C*.
DOI: https://doi.org/10.7554/eLife.38137.042
**Source data 4.** Data for the intensity profile plots of Bnl:GFP and *yan* (αYan) in *Figure 6D*.
DOI: https://doi.org/10.7554/eLife.38137.043
**Source data 5.** Numerical data of showing effects of *bnl:gfp* GOF clones in the ASP stalk in *Figure 6K,L*; *Figure 6—figure supplement 1D*.
DOI: https://doi.org/10.7554/eLife.38137.044
**Source data 6.** Numerical data of clonal analysis showing reciprocal inhibition of *pntP1* and *cut* in the ASP in *Figure 6M–P*.
DOI: https://doi.org/10.7554/eLife.38137.045
**Source data 7.** Data for the additional intensity profile plots of Bnl:GFP and *sty, pntP1, cut,* and *yan* in *Figure 6—figure supplement 2A–H*
DOI: https://doi.org/10.7554/eLife.38137.046
**Figure supplement 1.** Bnl:GFP acts as a morphogen.
DOI: https://doi.org/10.7554/eLife.38137.038
**Figure supplement 2.** Bnl:GFP acts as a morphogen.
DOI: https://doi.org/10.7554/eLife.38137.039

Notably, although PntP1 is a known inducer of *btl*, *pntP1* GOF/LOF clones showed only cell-autonomous effects on *cut*. The *btl* GOF clones in the ASP showed a similar autonomous signaling effect (*Figure 6—figure supplement 1E*). In contrast, each Bnl:GFP GOF clone in the ASP induced non-autonomous effects and organized the surrounding cells to form a new branch (N = 35; *Figure 6K, L*). All these results provided evidence that Bnl acts as a classical morphogen to induce *sty* at a high level and *pntP1*, *cut*, and *yan* gradually at lower levels in the ASP (*Figure 6E,F*).

## Target genes of Bnl signaling differentially feedback regulate cytoneme numbers

To examine whether the different target genes of Bnl signaling may regulate cytoneme formation, we focused on *pntP1* and *cut*, the two transcription factors that formed counteracting gradients from two opposing poles of the ASP and antagonized each other's expression (*Figures 6B,C,E,M–P*). When we induced ectopic *cut* GOF clones in the distal tip/mid-ASP regions, the mutant cells produced significantly fewer long and oriented cytonemes compared to the *wt* clones at an equivalent position (*Figure 7A,A'* versus *Figure 5A'',C*). This cytoneme deficiency in the *cut* GOF clones correlated with the autonomous reduction of Bnl:GFP uptake in the mutant clones from the mid/tip of the ASP (*Figure 7A–B'*). On the other hand, the orientation and number of cytonemes from the ASP stalk cells, which generally had high levels of Cut and produced randomly oriented short cytonemes, remained unaffected by *cut* GOF (*Figure 7A'*). Thus, high levels of Cut negatively regulate the formation of cytonemes that orient specifically toward the *bnl*-source. In addition to this finding, Cut was also known to be a suppressor of *btl* in the larval spiracular branch (*Pitsouli and Perrimon, 2013*). Therefore, high levels of Cut elicit negative feedback on Bnl signaling by suppressing both

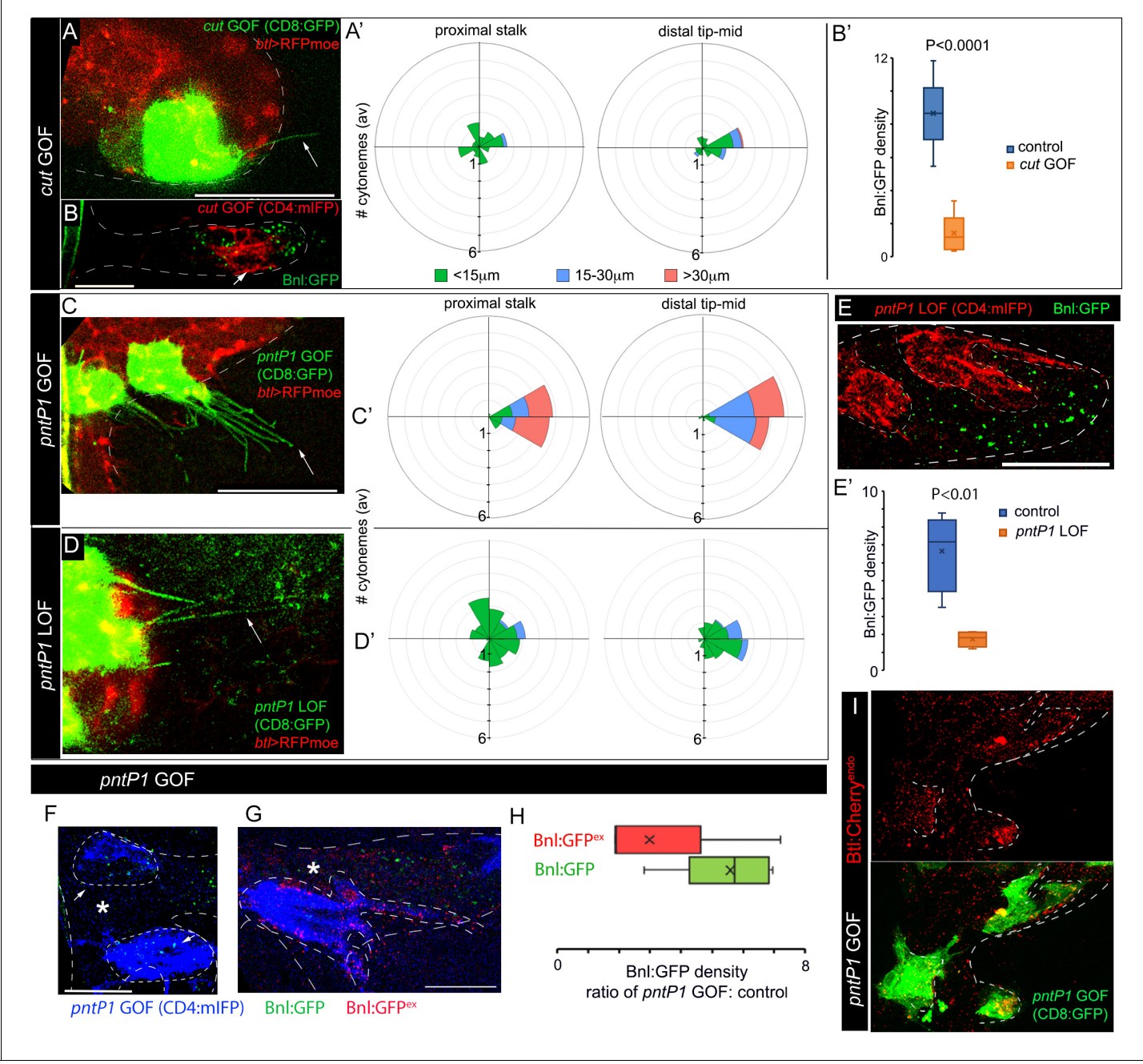

**Figure 7.** Positive and negative feedback regulations of cytoneme formation by differential levels of Bnl signaling. (**A,A'**) CD8:GFP-marked *cut*-GOF clones at the ASP tip suppressed long oriented cytoneme (arrow) formation; A', rose plots showing number and orientation of cytonemes of different lengths from distal tip (N = 12) and proximal stalk (N = 6) regions of the ASP. (**B,B'**) A *cut*-GOF clone (red, CD4:mIFP) suppressed Bnl:GFP uptake cell-autonomously; B', A graph comparing Bnl:GFP concentration in *cut*-GOF clones and corresponding neighboring control area in the ASP (N = 6); p value, two-tailed t-test. (**C,C'**) *pntP1*-GOF clones induced long oriented cytonemes (arrow) at the ASP stalk; C', rose plots showing number and orientation of cytonemes of different lengths from proximal (N = 5) and distal tip (N = 8) clones in the ASP. (**D,D'**) *pntP1*-LOF clones suppressed long oriented cytonemes (arrow) at the ASP tip; D', rose plots showing number and orientation of cytonemes of different lengths from proximal stalk (N = 6) and distal tip (N = 11) clones in the ASP. (**A,A',C,C',D,D'**) see control in *Figure 5A–C*. (**E,E'**) A CD4:mIFP-marked *pntP1*-LOF clone at the ASP tip suppressed Bnl:GFP uptake; E', plot comparing Bnl:GFP uptake in the *pntP1* LOF clone and corresponding neighboring control area in the ASP (N = 4). (**F**) *pnt-P1* GOF clones in the ASP stalk showed higher levels of Bnl:GFP in the clones compared to the WT neighboring area. (**G**) αGFP EIF showed high Bnl:GFP$^{ex}$ on cytonemes and the clonal cell body on *pnt-P1* GOF clones in the ASP stalk; (**F,G**) *, *wt* neighboring area. (**H**) A graph showing fold difference of Bnl:GFP and Bnl:GFP$^{ex}$ density inside to outside (WT control area) of the *pntP1* GOF clones (N = 5 each). (**I**) *pntP1* GOF clones induced *btl:cherry$^{endo}$* expression and polarized graded organization of Btl:Cherry molecules. Genotypes: (**A,A',C-D'**) *hs-FLP/+; UAS-mCD8:GFP/+;*

*Figure 7 continued on next page*

*Figure 7 continued*

*btl>y⁺>Gal4, btl-mRFP1moe/UAS-cut or -pntP1 or -pntRNAi; (B,B',E-H) hs-FLP/+; btl>y⁺>Gal4/+; UAS-CD4:mIFP, bnl:gfp^endo/UAS-cut or -pntRNAi or pntP1; (I) hs-FLP/+; btl>y⁺>Gal4/+; btl:cherry^endo/UAS-pntP1, UAS-mCD8:GFP.* Scale bars, 30 µm.

DOI: https://doi.org/10.7554/eLife.38137.047

The following source data and figure supplement are available for figure 7:

**Source data 1.** Numerical data for the number of cytonemes oriented in different directions from various clones in *Figure 7A',C',D'*.

DOI: https://doi.org/10.7554/eLife.38137.049

**Source data 2.** Numerical data for measuring the Bnl:GFP concentration in *cut*-GOF clones and the neighboring control area in *Figure 7B'*.

DOI: https://doi.org/10.7554/eLife.38137.050

**Source data 3.** Numerical data for measuring the Bnl:GFP concentration in *pntP1*-LOF clones and the neighboring control area in *Figure 7E'*.

DOI: https://doi.org/10.7554/eLife.38137.051

**Source data 4.** Numerical data for comparing the Bnl:GFP and Bnl:GFP^ex concentrations in *pntP1*-GOF clones and WT neighbors in *Figure 7H*.

DOI: https://doi.org/10.7554/eLife.38137.052

**Source data 5.** Numerical data for the number of cytonemes oriented in different directions from the *yan* GOF clones in *Figure 7—figure supplement 1A',B'*.

DOI: https://doi.org/10.7554/eLife.38137.053

**Figure supplement 1.** Yan negatively feedback regulates cytoneme numbers.

DOI: https://doi.org/10.7554/eLife.38137.048

the Bnl-receiving cytonemes and Btl synthesis (*Figure 8A,A'*). A similar inhibitory effect on cytonemes was observed for Yan, which was a known antagonist for PntP1 and embryonic tracheal filopodia formation (*Okenve-Ramos and Llimargas, 2014*) (*Figure 7—supplement figure 1A-B'; Figure 8A'*). Collectively, these results showed that *cut* and *yan*, the two genes that are expressed in the ASP stalk in response to lower Bnl levels, suppress cytoneme-formation and Btl expression.

Generally, high-to-mid levels of Bnl signaling induced *pntP1* in the distal tip-mid ASP region (*Figure 6B,E*). When ectopic *pntP1* GOF clones were generated in the ASP stalk, the mutant cells extended significantly more long (>30 µm) cytonemes oriented toward the underlying disc *bnl*-source than the WT clones (*Figure 7C,C'*, compared to *Figure 5A,C*). In contrast, *pntP1*-LOF clones at the ASP tip extended a significantly fewer number of long oriented cytonemes from the tip cells than the WT clones (*Figure 7D,D''*, compared to *Figure 5A'',C*). Loss of cytonemes in the *pntP1* LOF clones correlated with the suppression of Bnl:GFP uptake in the mutant ASP cells, and an increase in the number of cytonemes in the *pntP1* GOF cells increased the levels of Bnl:GFP in them (*Figure 7E–H*). An αGFP EIF assay revealed that the long cytonemes projecting from the *pntP1* GOF clones in the ASP stalk were enriched in EIF-stained Bnl:GFP^ex, suggesting that the cells received signal by extending cytonemes (*Figure 7G,H*). In addition, PntP1 was known to induce *btl* transcription in embryonic trachea (*Ohshiro et al., 2002*). When we generated small (1–4 cells) *pntP1* GOF clones in the stalk of the ASP in *btl:cherry^endo* larvae, the PntP1 overexpression increased the levels of Btl:Cherry in the clones. Strikingly, the newly synthesized Btl:Cherry molecules also formed local concentration gradients within each of the *pntP1* GOF clones (*Figure 7I*). These results showed that PntP1 elicits positive-feedback on Btl synthesis and Bnl-receiving cytoneme production (*Figure 8A'*). Interestingly, PntP1 and Cut (*Figure 6M–P*) or PntP1 and Yan (*Ohshiro et al., 2002*) can reciprocally inhibit each other's expression. Therefore, a ratio of the levels of PntP1 to Cut or to Yan could determine the number of cytonemes and Btl receptors in a cell. Based on our results, we propose that the counteracting activities of PntP1 and Cut/Yan and their expression from two opposite poles in response to the high-to-low Bnl levels in the ASP establish a steady-state gradient of numbers of Btl-containing cytonemes. Signaling through these cytonemes would self-generate the shape of a robust signal and signaling gradient (*Figure 8A'*).

## Discussion

### Dynamic, recipient tissue-specific shapes of an FGF morphogen gradient

Prior to this study, branching morphogenesis of tissues such as vertebrate lung or vasculature was proposed to be induced by exogenous, diffusible signal gradients. For instance, a mesenchymal

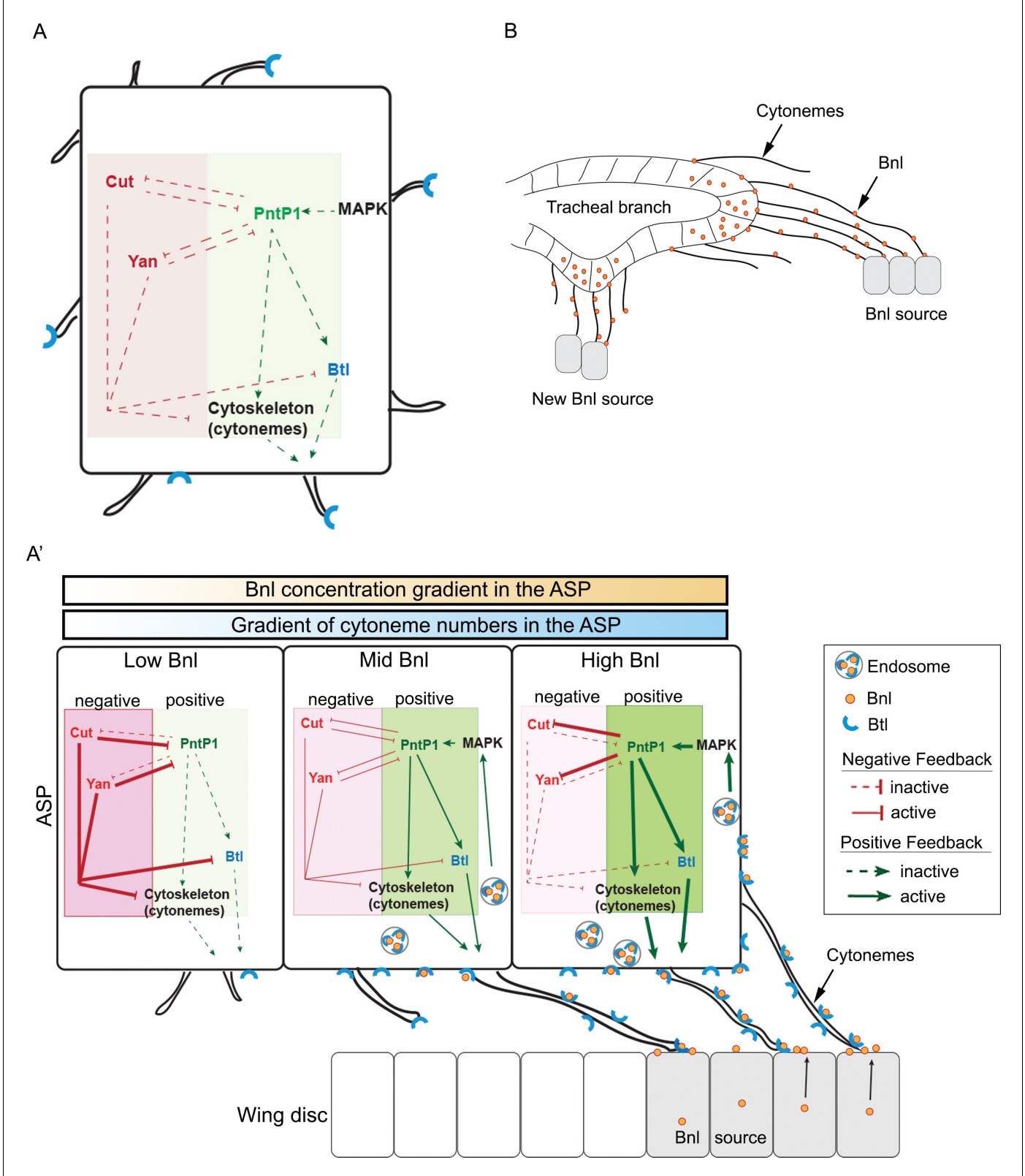

**Figure 8.** Feedback mechanisms regulating cytoneme-mediated Bnl gradient formation. (**A**) A hypothetical state of an isolated tracheal epithelial cell projecting cytonemes in random orientation prior to its establishment of cytoneme-mediated contact with a *bnl* source; Each recipient cell is endowed with a PntP1-dependent positive (green area) and Cut/Yan-dependent negative (red area) feedback, controlling the number of cytonemes and Btl (blue). (**A'**) Gradient formation is initiated when one or few tracheal cell(s) establish contact with a Bnl source. The ASP cells projecting Btl-containing

*Figure 8 continued on next page*

*Figure 8 continued*

cytonemes contact the *bnl*-source and directly receive Bnl signal (orange circles). Receptor-bound Bnl ligands move along the cytoneme surface and are endocytosed in the ASP cell to activate concentration-dependent gene activities: PntP1 at high-moderate levels, and Cut and Yan at gradually lower levels. PntP1 elicits positive feedback while Cut and Yan elicit negative feedback on production of Btl and cytonemes in tracheal cells, creating a steady-state graded pattern of Bnl-receiving cytonemes. Signaling through these cytonemes generates and reinforces the Bnl gradient. The positive (PntP1) and negative (Cut/Yan) feedback also inhibit each other, and thereby can shape the gradient following recipient tissue-specific morphology. (B) When few tracheal cells from a pre-existing branch establish contacts with a new proximal source, this mechanism can initiate a new local gradient and a new epithelial branch organization.

DOI: https://doi.org/10.7554/eLife.38137.054

FGF10 gradient induces branching in the vertebrate lung, and a Vascular Endothelial Growth Factor/ VEGF gradient induces vascular sprouting (*Affolter et al., 2009*; *Spurlin and Nelson, 2017*). Although the paracrine functions of these signals were well characterized, our demonstration and detailed characterization of the Bnl gradient represents the first example of an inductive gradient for branching morphogenesis. In contrast to the traditional idea of a preexisting diffusible gradient, we showed that the shapes of the inductive Bnl gradient are dynamic, adaptable, and form only in a receptor-bound state within the recipient tissue. Moreover, we showed that the Bnl gradient self-generates its variable recipient branch-specific shapes by regulating its cytoneme-mediated transport. Self-generation of tissue-specific gradients may be a common and essential strategy for sustaining the robust, yet variable gradient shapes in dynamically migrating tissues. Consistent with this view, the chemokine Cxcl12a also forms a self-generating gradient to regulate dynamic migration of the Zebrafish lateral line (*Donà et al., 2013*). Notably, inductive signals such as Bnl or VEGF were previously thought to induce chemotactic migration by inducing binary signaling responses in the recipient cells (*Ochoa-Espinosa and Affolter, 2012*). Therefore, identification of an organizer-like morphogenetic role of Bnl brings a critical shift to our understanding of branching morphogenesis.

## Cytoneme-mediated, tissue-specific Bnl dispersion

Several earlier reports, including biophysical analyses of vertebrate FGF8 and its distribution patterns in tissues, proposed a free or restricted diffusion-based dispersion mechanism for gradient formation (*Bökel and Brand, 2013*; *Toyoda et al., 2010*; *Yu et al., 2009*). An alternative model proposed that the FGF8 morphogen gradient is also formed by gradual decay of *fgf8* mRNA and its translation in a growing vertebrate embryo (*Dubrulle and Pourquié, 2004*). In contrast, all of our genetic analyses and direct imaging results show that the Bnl gradient is formed by cytoneme-mediated direct transport. An earlier study showed that ASP cells project Btl-containing cytonemes to establish contact with the disc Bnl source, and fail to induce pMAPK signaling without these cytoneme-mediated contacts (*Roy et al., 2014*). In this study we extended those initial findings. By generating endo-tagged Bnl:GFP and Btl:Cherry constructs expressed at physiological levels, utilizing an EIF assay to detect externalized Bnl molecules, and by applying high resolution visualization techniques, we obtained reliable and unbiased evidence for Bnl dispersion via cytonemes. We showed that ASP cytonemes receive Bnl molecules by establishing direct membrane synapses with the signal producing cells. Super-resolution imaging of membrane-marked ASP and source cells documented these synaptic sites and showed a selective enrichment of externalized Bnl molecules only at these signaling contacts (*Figure 3J*). At this point, we do not know why and how Bnl is released from producing cells only at cytoneme-source membrane contact sites. Understanding the mechanisms underlying this process is an important new direction for future investigations.

Antibody-based EIF analysis under detergent-free conditions and super-resolution imaging of endogenously expressed Bnl:GFP showed that at a steady state, a high concentration of the signal moves along the surface of the cytonemes in receptor-bound forms (*Figure 3C–H*). It is likely that receptor-bound Bnl ligands move along the surface of the cytonemes by an energy-dependent motor transport and are endocytosed when they reach the ASP cell body. Super-resolution imaging also revealed a previously uncharacterized sub-resolution distribution of Bnl:GFP nanopuncta (*Figure 3F–F''*; *Figure 3—supplement figure 2*), suggesting the existence of novel cellular and molecular events that regulate the cytoneme-mediated signaling process. Each of these individual cellular and molecular mechanisms would influence the levels of Bnl signaling in a cell and thereby

would contribute to the essential feedback mechanisms by which the cytoneme and the morphogen gradients are formed.

Contact-dependent Bnl exchange implies that the ASP cells can interpret the direction from where (and from whom) they receive the Bnl signal, in addition to its levels. We do not know how ASP cytonemes find the *bnl*-source and establish the signaling contacts. A recent study had shown an important regulatory role of the components of the extracellular matrix, which interact with the signal-specific cytonemes and stabilize their directionality (*Huang and Kornberg, 2016*). Another possible mechanism could be that the cells in the disc Bnl source extend Bnl-containing cytonemes to establish an initial contact with the recipient cytonemes to provide them the necessary pathfinding information. In fact, Hh is known to be delivered by the source cytonemes to the recipient cytonemes in *Drosophila* wing disc and vertebrate limb buds (*Chen et al., 2017*; *González-Méndez et al., 2017*; *Sanders et al., 2013*). However, the imaging conditions used in this study could not detect Bnl-containing source cytonemes. Future live imaging analyses are required to elucidate the mechanisms of cytoneme pathfinding.

## A self-regulatory mechanism for gradient formation

How morphogen gradients are produced in tissues is a long-standing central question. Based on our experimental results, we propose a model for the mechanism by which cytonemes can create and maintain a morphogen gradient (*Figure 8A'*). We uncovered that ASP epithelial cells extend a graded number of cytonemes that reach out and contact the disc *bnl* source to receive the signal (*Figure 5A–D''*). Recipient cells close to the source have many long polarized cytonemes, but their numbers gradually reduced as the distance from the source increases. Clonal GFP-reconstitution analyses suggested that the number of cytoneme-contacts an ASP cell makes is inversely proportional to its distance from the signal source (*Figure 5D–D''*). Since reception of Bnl is contact-dependent (*Figure 3I,I'*), these features of patterned cytoneme formation in the recipient ASP explain how cytoneme-mediated signal reception can generate the recipient tissue-specific shapes of the gradient. These observations led us to address a very important question: how are the cytoneme patterns developed and maintained in the tissue? Answering this question was expected to provide the basis by which cytonemes create and regulate gradient shapes and tissue patterns.

A key finding providing insight into this question came from our analysis of the spatial domains of targets of Bnl signaling in the ASP. One of the target genes, *pntP1*, is induced by high Bnl levels in the distal region of the ASP and positively regulates the formation of cytonemes and Btl synthesis (*Figure 6B,E*). On the other hand, surprisingly, the target gene *cut*, which is induced by gradually lower levels of Bnl in the ASP stalk region away from the source, negatively controls cytoneme formation (*Figure 6C,E; 7A,A'; 8A'*) and Btl synthesis (*Pitsouli and Perrimon, 2013*). Moreover, the transcription factors Cut and PntP1 feedback-inhibit each other's expression to maintain their zones of expression in response to Bnl levels (*Figure 6M–P*). The consequence is a regulatory loop that controls the generation and reinforcement of different numbers of cytonemes and levels of Bnl signaling, making the contours of the Bnl gradient robust, precisely tissue-specific, and self-sustaining.

The roles of these target genes might also explain how variable shapes of Bnl gradients are sculpted in coordination with the growth and development of recipient tissues. One current morphogen model proposes that the signal gradients are generated and scaled to the size of the recipient tissue by the ratio of two diffusible molecules, a morphogen and a contractor/expander, which emanate from opposite poles of the tissue and elicit non-autonomous global feedback to control each other's distribution (*Barkai and Shilo, 2009*; *Ben-Zvi et al., 2011*; *Restrepo and Basler, 2011*; *Shilo and Barkai, 2017*). Although Cut and PntP1 are intracellular transcription factors that function cell autonomously, their counteracting feedback activities, regulating where and when signal-specific cytonemes are made as a function of the levels of signal received, can self-generate gradients adopting any recipient-specific shape (*Figures 1F; 8A'*).

This mechanism implies an interesting systemic feedback where the gradient shape is a result of the growth and patterning of the recipient tissue, which it controls. However, this mechanism also implies that to initiate such a self-sustaining gradient, initial physical contact of a part of the recipient epithelium with the signal source is required. In fact, each tracheal cell is capable of inducing either PntP1 or Cut/Yan expression irrespective of its position within the recipient epithelium (*Figure 8A*). Thus, an ectopic new *bnl* source near a pre-existing tracheal branch can initiate a local gradient of cytoneme contacts leading to polarized signal and signaling gradient formation as well as neo-

branch organization (*Figure 8B*). In agreement with this view, spatiotemporal expression of Bnl surrounding the embryonic tracheal placode is known to induce new branches (*Sutherland et al., 1996*). Moreover, if the Bnl response genes are not the same or are not regulated in the same manner at all sites and at all times, different patterns of cytoneme formation can be generated leading to changes in spatial patterns of signal distribution and tissue morphology. Given the commonality of fundamental signaling events mediated by conserved signaling proteins, feedback regulation of cytoneme-mediated transport may offer an explanation for why signal gradients are so precise, yet adaptable and for how diverse tissue morphologies can result from just one signal transduction pathway.

# Materials and methods

## Key resources table

| Reagent type (species) or Resource | Designation | Source or reference | Identifiers | Additional information |
|---|---|---|---|---|
| Antibody | mouse anti-Discs large | DSHB | DSHB:4F3 RRID: AB_528203 | IHC (1:100) |
| Antibody | rabbit anti-dpERK (Phospho-p44/42 MAPK (Erk1/2) (Thr202/Tyr204) Rabbit mAb | Cell signaling Technology | Cell signaling Technology:4370 RRID:AB_11207064 | IHC (1:100) |
| Antibody | mouse anti-Cut | DSHB | DSHB:2B10 RRID:AB_528186 | IHC (1:50) |
| Antibody | mouse anti-beta-galactosidase | DSHB | DSHB:40-1a RRID:AB_528100 | IHC (1:50) |
| Antibody | rabbit anti-Rab5 | Abcam | Abcam:ab31261 RRID:AB_882240 | IHC (1:10000) |
| Antibody | rabbit anti-Rab7 | (*Tanaka and Nakamura, 2008*) | N/A | IHC (1:3000) |
| Antibody | rabbit anti-Rab11 | (*Tanaka and Nakamura, 2008*) | N/A | IHC (1:8000) |
| Antibody | rabbit anti-Lamp1 | Abcam | Abcam:ab30687 RRID:AB_775973 | IHC (1:10000) |
| Antibody | mouse anti-Yan | DSHB | DSHB:8B12H9 RRID:AB_531807 | IHC (1:200) |
| Antibody | rabbit anti-Bnl | This paper | N/A | IHC (1:1000), EIF (1:500) |
| Antibody | rabbit anti-GFP | Abcam | Abcam:ab6556 RRID:AB_305564 | EIF (1:3000) |
| Chemical compound, drug | Alexa Fluor 647 Phalloidin | Thermo Fisher Scientific | Cat. #: A22287 RRID:AB_2620155 | (1:1000) |
| Genetic reagent (*Drosophila melanogaster*) | {nos-Cas9}ZH-2A | Bloomington Drosophila stock center | RRID:BDSC_54591 | |
| Genetic reagent (*Drosophila melanogaster*) | *UAS*-CD8:GFP | Bloomington Drosophila stock center | RRID:BDSC_5137 | |
| Genetic reagent (*Drosophila melanogaster*) | *UAS*-nlsGFP | Bloomington Drosophila stock center | RRID:BDSC_4776 | |
| Genetic reagent (*Drosophila melanogaster*) | *UAS*-CD8:RFP | Bloomington Drosophila stock center | RRID:BDSC_32218 | |

*Continued on next page*

*Continued*

| Reagent type (species) or Resource | Designation | Source or reference | Identifiers | Additional information |
|---|---|---|---|---|
| Genetic reagent (*Drosophila melanogaster*) | *UAS*-CD4:mIFP | Bloomington Drosophila stock center | RRID:BDSC_64182 | |
| Genetic reagent (*Drosophila melanogaster*) | *lexO-nsyb*:GFP[1-10], *UAS*-CD4:GFP[11] | Bloomington Drosophila stock center | RRID:BDSC_64315 | |
| Genetic reagent (*Drosophila melanogaster*) | *UAS*-Btl[DN] | (*Reichman-Fried and Shilo, 1995*) | N/A | |
| Genetic reagent (*Drosophila melanogaster*) | *UAS*-bnlRNAi | Bloomington Drosophila stock center | RRID:BDSC_34572 | |
| Genetic reagent (*Drosophila melanogaster*) | *UAS*-pntRNAi | Bloomington Drosophila stock center | RRID:BDSC_35038 | |
| Genetic reagent (*Drosophila melanogaster*) | *UAS*-cutRNAi | Bloomington Drosophila stock center | RRID:BDSC_33967 | |
| Genetic reagent (*Drosophila melanogaster*) | *UAS*-btlRNAi | Bloomington Drosophila stock center | RRID:BDSC_40871 | |
| Genetic reagent (*Drosophila melanogaster*) | *UAS*-diaRNAi | Bloomington Drosophila stock center | RRID:BDSC_33424 | |
| Genetic reagent (*Drosophila melanogaster*) | *UAS*-nrgRNAi | Bloomington Drosophila stock center | RRID:BDSC_37496 | |
| Genetic reagent (*Drosophila melanogaster*) | *UAS*-snRNAi | Bloomington Drosophila stock center | RRID:BDSC_42615 | |
| Genetic reagent (*Drosophila melanogaster*) | *pnt-lacZ*[07825] | Bloomington Drosophila stock center | RRID:BDSC_11724 | |
| Genetic reagent (*Drosophila melanogaster*) | *sty-lacZ* | Bloomington Drosophila stock center | RRID:BDSC_11735 | |
| Genetic reagent (*Drosophila melanogaster*) | *bnl-LexA* | (*Du et al., 2017*) | | |
| Genetic reagent (*Drosophila melanogaster*) | *bnl-Gal4* | Bloomington Drosophila stock center | RRID:BDSC_112825 | |
| Genetic reagent (*Drosophila melanogaster*) | *btl-Gal4* | (*Sato and Kornberg, 2002*) | N/A | |
| Genetic reagent (*Drosophila melanogaster*) | *btl-LHG* | (*Roy et al., 2014*) | N/A | |
| Genetic reagent (*Drosophila melanogaster*) | *UAS*-Btl | (*Roy et al., 2011a*) | N/A | |
| Genetic reagent (*Drosophila melanogaster*) | *UAS*-CD8:Cherry | (*Roy et al., 2011a*) | N/A | |

*Continued on next page*

*Continued*

| Reagent type (species) or Resource | Designation | Source or reference | Identifiers | Additional information |
|---|---|---|---|---|
| Genetic reagent (*Drosophila melanogaster*) | *lexO*-CD2:GFP | (*Yagi et al., 2010*) | N/A | |
| Genetic reagent (*Drosophila melanogaster*) | hs-FLP; btl>y+>Gal4, btl-mRFP1moe | (*Cabernard and Affolter, 2005*) | N/A | |
| Genetic reagent (*Drosophila melanogaster*) | hs-FLP; btl>y+>Gal4; btl-mRFP1moe | (*Cabernard and Affolter, 2005*) | N/A | |
| Genetic reagent (*Drosophila melanogaster*) | bnl:gfp^endo | This paper | N/A | Functional genomic knock-in allele by CRISPR |
| Genetic reagent (*Drosophila melanogaster*) | btl:cherry^endo | This paper | N/A | Functional genomic knock-in allele by CRISPR |
| Genetic reagent (*Drosophila melanogaster*) | UAS-secGFP | This paper | N/A | Secreted GFP overexpression construct |
| Sequence-based reagent | 5'-GTCGGGG CCAATCGCGTCAAGCA-3' | This paper | N/A | Guide RNA-1 for bnl:gfp^endo |
| Sequence-based reagent | 5'-AAACTGCT TGACGCGATTGGCCC-3' | This paper | N/A | Guide RNA-1 for bnl:gfp^endo |
| Sequence-based reagent | 5'-GTCGATAT TAGCAGTAGCCTTAG-3' | This paper | N/A | Guide RNA-2 for bnl:gfp^endo |
| Sequence-based reagent | 5'-AAACCTAA GGCTACTGCTAATAT-3' | This paper | N/A | Guide RNA-2 for bnl:gfp^endo |
| Sequence-based reagent | 5'-GTCGCAT CACGGAGACGGTGCCGC-3' | This paper | N/A | Guide RNA-3 for bnl:gfp^endo |
| Sequence-based reagent | 5'-AAACGCGG CACCGTCTCCGTGATG-3' | This paper | N/A | Guide RNA-3 for bnl:gfp^endo |
| Sequence-based reagent | 5'-GTCGAGG TGTACTGATATCTAAG-3' | This paper | N/A | Guide RNA-1 for btl:cherry^endo |
| Sequence-based reagent | 5'-AAACCTTA GATATCAGTACACCT-3' | This paper | N/A | Guide RNA-1 for btl:cherry^endo |
| Sequence-based reagent | 5'-GTCGCGG CATCGAAAGGTCCAGAT-3' | This paper | N/A | Guide RNA-2 for btl:cherry^endo |
| Sequence-based reagent | 5'-AAACATCT GGACCTTTCGATGCCG-3' | This paper | N/A | Guide RNA-2 for btl:cherry^endo |
| Sequence-based reagent | 5'-CGTATGGG ATTCCGATTGTGTG-3' | This paper | N/A | Primer to amplify N fragment of HDR donor for bnl:gfp^endo |
| Sequence-based reagent | 5'-CAGCTCCT CGCCCTTGGACATAGTG TTGCTGCTGCAATGTGGCGG-3' | This paper | N/A | Primer to amplify N fragment of HDR donor for bnl:gfp^endo |
| Sequence-based reagent | 5'-CCGCCACA TTGCAGCAGCAACACTA TGTCCAAGGGCGAGGAGCT-3' | This paper | N/A | Primer to amplify mid fragment of HDR donor for bnl:gfp^endo |
| Sequence-based reagent | 5'-CTGCTGA TGCTGCTGCTGCTGCCAC TCTTGTACAGCTCATC CATGCCCAG-3' | This paper | N/A | Primer to amplify mid fragment of HDR donor for bnl:gfp^endo |
| Sequence-based reagent | 5'-CTGGGCA TGGATGAGCTGTACAAG AGTGGCAGCAGCAGCA GCATCAGCAG-3' | This paper | N/A | Primer to amplify C fragment of HDR donor for bnl:gfp^endo |

*Continued on next page*

*Continued*

| Reagent type (species) or Resource | Designation | Source or reference | Identifiers | Additional information |
|---|---|---|---|---|
| Sequence-based reagent | 5'-GGCTTGA GAGGTTCTTATAAAA TACTCGAG-3' | This paper | N/A | Primer to amplify C fragment of HDR donor for bnl:gfp[endo] |
| Sequence-based reagent | 5'-CTCAACTT CACCGTGACGAATGAC-3' | This paper | N/A | Primer to amplify N fragment of HDR donor for btl:cherry[endo] |
| Sequence-based reagent | 5'-GTTTCTCC ATGCGCTGACCCGT AATCAG-3' | This paper | N/A | Primer to amplify N fragment of HDR donor for btl:cherry[endo] |
| Sequence-based reagent | 5'-CTGATTAC GGGTCAGCGCATG GAGAAAC-3' | This paper | N/A | Primer to amplify mid fragment of HDR donor for btl:cherry[endo] |
| Sequence-based reagent | 5'-GGAATTCT TTTTGGTCTCCTTAT ACTACGAA-3' | This paper | N/A | Primer to amplify mid fragment of HDR donor for btl:cherry[endo] |
| Sequence-based reagent | 5'-TTCGTAGT ATAAGGAGACCAAA AAGAATTCC-3' | This paper | N/A | Primer to amplify C fragment of HDR donor for btl:cherry[endo] |
| Sequence-based reagent | 5'-GGTTCCTC TTCCATCCAAGGTTG-3' | This paper | N/A | Primer to amplify C fragment of HDR donor for btl:cherry[endo] |
| Sequence-based reagent | 5'-TTTTGGG GCCAATCGTGTGAAGC ACGGCGTGCGG-3' | This paper | N/A | Primer to introduce synonymous mutation at bnl-gRNA-1 recognition site in HDR donor |
| Sequence-based reagent | 5'-CCGCACG CCGTGCTTCACACGAT TGGCCCCAAAA-3' | This paper | N/A | Primer to introduce synonymous mutation at bnl-gRNA-1 recognition site in HDR donor |
| Sequence-based reagent | 5'-CCATATTA GCAGTAGTCTGAGCGG TAGCAGTAAC-3' | This paper | N/A | Primer to introduce synonymous mutation at bnl-gRNA-2 recognition site in HDR donor |
| Sequence-based reagent | 5'-GTTACTGC TACCGCTCAGACTACT GCTAATATGG-3' | This paper | N/A | Primer to introduce synonymous mutation at bnl-gRNA-2 recognition site in HDR donor |
| Sequence-based reagent | 5'-GGAGACG GTGCCGCAAGAGCGG GTCGAGCAG-3' | This paper | N/A | Primer to introduce synonymous mutation at bnl-gRNA-3 recognition site in HDR donor |
| Sequence-based reagent | 5'-CTGCTCG ACCCGCTCTTGCGGC ACCGTCTCC-3' | This paper | N/A | Primer to introduce synonymous mutation at bnl-gRNA-3 recognition site in HDR donor |
| Sequence-based reagent | 5'-CCGGGAA ACGTCCCCGCTGAGGT ATCAGTACACCTATAAG-3' | This paper | N/A | Primer to introduce synonymous mutation at btl-gRNA-1 recognition site in HDR donor |
| Sequence-based reagent | 5'-CTTATAGG TGTACTGATACCTCAG CGGGGACGTTTCCCGG-3' | This paper | N/A | Primer to introduce synonymous mutation at btl-gRNA-1 recognition site in HDR donor |
| sequence-based reagent | 5'-GTAGCAAT CCAAACGATGCGTAT CTGGACCTTTCGATGC-3' | This paper | N/A | Primer to introduce synonymous mutation at btl-gRNA-2 recognition site in HDR donor |

*Continued on next page*

*Continued*

| Reagent type (species) or Resource | Designation | Source or reference | Identifiers | Additional information |
|---|---|---|---|---|
| Sequence-based reagent | 5'-GCATCGA AAGGTCCAGATACGCA TCGTTTGGATTGCTAC-3' | This paper | N/A | Primer to introduce synonymous mutation at btl-gRNA-2 recognition site in HDR donor |
| Sequence-based reagent | 5'-GTCCTGTT TAGGGGCGATAAGTGG-3' | This paper | N/A | Primer for bnl:gfp$^{endo}$ HDR screening and sequencing |
| Sequence-based reagent | 5'-GTGTTGCG TAAGGTTAGGGCTTCG-3' | This paper | N/A | Primer for bnl:gfp$^{endo}$ HDR screening and sequencing |
| Sequence-based reagent | 5'-GAAGCAG CACGATTTCTTCAAGAGCG-3' | This paper | N/A | Primer for bnl:gfp$^{endo}$ HDR screening and sequencing |
| Sequence-based reagent | 5'-CGCTCTTG AAGAAATCGTGCTGCTTC-3' | This paper | N/A | Primer for bnl:gfp$^{endo}$ HDR screening and sequencing |
| Sequence-based reagent | 5'-CGCCAGC CAGGCAAT-3' | This paper | N/A | Primer for bnl:gfp$^{endo}$ HDR screening and sequencing |
| Sequence-based reagent | 5'-GTCCTCAA GAATGCCTCCTTGGAC-3' | This paper | N/A | Primer for btl:cherry$^{endo}$ HDR screening and sequencing |
| Sequence-based reagent | 5'-GTCTATGA TACCTCTGACAGCTTC-3' | This paper | N/A | Primer for btl:cherry$^{endo}$ HDR screening and sequencing |
| Sequence-based reagent | 5'-CTTCCCC GAGGGCTTCAAGTG-3' | This paper | N/A | Primer for btl:cherry$^{endo}$ HDR screening and sequencing |
| Sequence-based reagent | 5'-CACTTGAA GCCCTCGGGGAAG-3' | This paper | N/A | Primer for btl:cherry$^{endo}$ HDR screening and sequencing |
| Sequence-based reagent | 5'-GTTTCTCC ATGCGCTGACCCG TAATCAG-3' | This paper | N/A | Primer for btl:cherry$^{endo}$ HDR screening and sequencing |
| Sequence-based reagent | 5'-AATTCGAG CTCGGTACAGATCTA TGCGAAGAAACCTGCGC-3' | This paper | N/A | Primer for UAS-SP$_{Bnl}$-sfGFP cloning |
| Sequence-based reagent | 5'-CCTCGCCC TTGGACATCATCGCA GATACAAGGCCCC-3' | This paper | N/A | Primer for UAS-SP$_{Bnl}$-sfGFP cloning |
| Sequence-based reagent | 5'-GGCCTTGT ATCTGCGATGATGTC CAAGGGCGAGGAG-3' | This paper | N/A | Primer for UAS-SP$_{Bnl}$-sfGFP cloning |
| Sequence-based reagent | 5'-GCCAAGC TTGCATGCCGGTACCT TACTTGTACAGCTC ATCCATGCCC-3' | This paper | N/A | Primer for UAS-SP$_{Bnl}$-sfGFP cloning |
| Recombinant DNA reagent | pCR4Blunt-TOPO | Thermo Fisher Scientific | N/A | |
| Recombinant DNA reagent | pUC19 | Addgene | Addgene:50005 | |
| Recombinant DNA reagent | pUAST | Drosophila Genomics Resource center | DGRC:1000 | |
| Recombinant DNA reagent | pCFD3 | (*Port et al., 2014*) | N/A | |

*Continued on next page*

*Continued*

| Reagent type (species) or Resource | Designation | Source or reference | Identifiers | Additional information |
|---|---|---|---|---|
| Recombinant DNA reagent | pCFD3-bnl_gRNA-1 | This paper | N/A | Plasmid for generation of bnl_gRNA-1 flies |
| Recombinant DNA reagent | pCFD3-bnl_gRNA-2 | This paper | N/A | Plasmid for generation of bnl_gRNA-2 flies |
| Recombinant DNA reagent | pCFD3-bnl_gRNA-3 | This paper | N/A | Plasmid for generation of bnl_gRNA-3 flies |
| Recombinant DNA reagent | pCFD3-btl_gRNA-1 | This paper | N/A | Plasmid for generation of btl_gRNA-1 flies |
| Recombinant DNA reagent | pCFD3-btl_gRNA-2 | This paper | N/A | Plasmid for generation of btl_gRNA-2 flies |
| Recombinant DNA reagent | pCR4Blunt-TOPO-bnl:GFP-1 | This paper | N/A | HDR donor for making bnl:gfp$^{endo}$ flies |
| Recombinant DNA reagent | pCR4Blunt-TOPO-bnl:GFP-2 | This paper | N/A | HDR donor for making bnl:gfp$^{endo}$ flies |
| Recombinant DNA reagent | pCR4Blunt-TOPO-bnl:GFP-3 | This paper | N/A | HDR donor for making bnl:gfp$^{endo}$ flies |
| Recombinant DNA reagent | pCR4Blunt-TOPO-btl:Cherry-1 | This paper | N/A | HDR donor for making btl:cherry$^{endo}$ flies |
| Recombinant DNA reagent | pCR4Blunt-TOPO-btl:Cherry-2 | This paper | N/A | HDR donor for making btl:cherry$^{endo}$ flies |
| Recombinant DNA reagent | pUAST-SP$_{Bnl}$-sfGFP | This paper | N/A | Plasmid for generation of UAS-secGFP flies |
| Software, algorithm | Fiji (ImageJ v2.0) | (*Schindelin et al., 2012*) | RRID:SCR_002285 | |
| Software, algorithm | Adobe Photoshop | Adobe | RRID:SCR_014199 | |
| Software, algorithm | Microsoft Excel | Microsoft | RRID:SCR_016137 | |
| Software, algorithm | SnapGene | snapgene.com | RRID:SCR_015052 | |
| Software, algorithm | VassarStats | vassarstats.net | RRID:SCR_010263 | |
| Software, algorithm | R | r-project.org | RRID:SCR_001905 | |

## CRISPR/Cas9-based genome-editing

The *bnl:gfp$^{endo}$* and *btl:cherry$^{endo}$* alleles were generated following CRISPR/Cas9 based genome editing method described earlier (*Du et al., 2017*) (*Figures 1B* and *2A*; *Figure 1—figure supplement 1A–C*; *Figure 2—figure supplement 1A–C*). The *bnl:gfp$^{endo}$* knock-in allele expressed a chimeric Bnl:GFP protein containing a superfolder-GFP (sfGFP) fused in-frame at ~53 amino acid downstream to the conserved FGF domain (247 to 376 amino-acid) (*Figure 1B*). Due to its high solubility and efficient folding kinetics in extracellular environment (*Pédelacq et al., 2006*), we selected sfGFP for tagging Bnl. The *btl:cherry$^{endo}$* knock-in allele expressed Btl:Cherry containing in-frame C-terminal fusion of mCherry (*Figure 2A*).

Multiple gRNA binding sites with variable efficiency were selected for independent parallel genome-editing experiments to achieve the highest probability of obtaining the intended chimeras and identify the most consistent patterns of Bnl:GFP and Btl:Cherry distribution in tissues among various lines. The gRNA expression constructs were cloned by ligating the annealed complementary oligos (Key resources table) as described in (*Du et al., 2017*). The gRNAs (PAM sequence underlined) used for generating *bnl:gfp$^{endo}$* and *btl:cherry$^{endo}$* were:

BnlGFP_gRNA-1: GGGCCAATCGCGTCAAGCA<u>CGG</u>
BnlGFP_gRNA-2: ATATTAGCAGTAGCCTTAG<u>CGG</u>
BnlGFP_gRNA-3: CATCACGGAGACGGTGCCGC<u>AGG</u>
BtlCherry_gRNA-1: AGGTGTACTGATATCTAAG<u>CGG</u>
BtlCherry_gRNA-2: CGGCATCGAAAGGTCCAGAT<u>AGG</u>

Replacement donors, *pDonor-bnl:GFP*, and *pDonor-btl:Cherry* were designed and generated following (*Du et al., 2017*). For efficient 'Homology-Directed Repair' (HDR), these constructs contained the tag sequence flanked by ~1.5 to 1.7 kb long gene-specific 5'- and 3'- homology arms (*Figure 1—figure supplement 1A*; *Figure 2—figure supplement 1A*; Key resources table). Both 5'- and 3'-homology arms were PCR-amplified from the genomic DNA from *nos-cas9* parent fly, sequence verified, and were fused together with the *sfgfp* or *mCherry* sequence by overlap extension PCR. To avoid retargeting of the gRNA to the engineered locus, synonymous mutations were introduced into or near the PAM sequence of the homology arms via the primers used for their amplification (Key resources table). The clones were fully sequenced before being used for genomic replacement.

Transgenic flies harboring gRNA vectors at the *attP40* landing site (*y[1] v[1] nos-phiC31; attP40* host) were generated following (*Kondo and Ueda, 2013*). Males harboring a specific *U6:3-gRNA* were crossed to *nos*-Cas9 (*nos-Cas9* ZH2A, BL# 54591) females to obtain *nos-cas9/+; U6:3-gRNA/+*embryos. For each genome-editing experiment, the corresponding replacement donor plasmid was injected into the germline of these embryos. A step-wise crossing strategy, as described earlier (*Du et al., 2017*), was followed to obtain the G[0]-F2 progenies and establish individual fly lines for screening. The correct 'ends-out' HDR was screened by a PCR based strategy (*Du et al., 2017*), followed by the analyses of tissue-specific expression patterns of the tagged-genes under a confocal microscope (*Figure 1—figure supplement 1A–C*; *Figure 2—figure supplement 1A–C*).

Amplification products obtained using PCR primers fwd1-rev2 and fwd2-rev1 confirmed knock-in positive lines, whereas amplification by the M13F and rev3 primers indicated unintended 'ends-in' HDR. Finally, a correct sized PCR product obtained using primers fwd1-rev1 confirmed the ends-out HDR (*Figure 1—figure supplement 1A–B'''*; *Figure 2—figure supplement 1A–B'''*). All GFP or mCherry positive, sequence verified lines were outcrossed to remove unintended mutations (if any), and any putative non-specific off-target sites were further sequence confirmed to be free of genome-editing. The homozygous *btl*:cherry[endo], and *bnl*:gfp[endo] fly lines with accurate genomic sequences had a normal tissue morphology, and expected expression/distribution patterns (*Figure 1C–D'*, *Figure 1—figure supplement 1C*; *Figure 2B*; *Figure 2—figure supplement 1C*). We used one BnlGFP_gRNA-3 (c26-9), and a Btl:Cherry_gRNA-1 (a21-10) line for all analyses.

## Molecular cloning

For secGFP, the N-terminal 32 amino-acid (96 nucleotides) long signal peptide sequence of Bnl was cloned in-frame upstream of a *sfgfp* sequence into pUC19 using Gibson Assembly (Key resources table). The product was sub-cloned into the BglII, Acc651 sites of the pUAST for P-element mediated germline transformation in *Drosophila*. A transgenic fly was generated that harbored an overexpression *UAS*-Bnl:GFP construct derived from *bnl* cDNA (DGRC).

## Genetic crosses

All crosses were incubated at 25°C unless specified otherwise.

For *nsyb*GRASP labeling of cytoneme contacts (*Figure 3I,I'*; *Videos 3* and *5*): *btl-Gal4,UAS-CD8:Cherry; bnl-LexA/TM6* virgin females were crossed to *lexO-nsyb:GFP[1-10],UAS-CD4:GFP[11]* males.

Flip-out clones of specific genotype were generated by 8–15 min 37°C heat shock of the early 3rd instar larvae followed by 25°C incubation until they reached the late 3rd instar stages (~24 hr).

1. To induce clones of a specific genotype in the ASP and check for Bnl:GFP uptake into the clones (*Figure 4A–H'*; *Figure 4—figure supplement 1A,B*; *Figure 7B,B',E-H*), *hs-FLP; btl>y +>Gal4; bnl:gfp[endo],UAS-CD4:mIFP* flies were crossed to either *w1118* (control) or *UAS-X* (X- *pntP1,pntRNAi, cut, btl, btlRNAi, diaRNAi, nrgRNAi,* or *snRNAi*). CD4:mIFP did not efficiently localize in cytonemes compared to CD8:GFP or CD8:RFP, CD8:Cherry or CherryCAAX.

2. To count the number of cytonemes emanating from the clones in the ASP (*Figure 5A–C; 7A,A',C-D'*; *Figure 7—figure supplement 1A–B'*), *hs-FLP; UAS-mCD8:GFP; btl>y +>Gal4, btl-mRFP1moe* flies were crossed to either *w1118* (control), or *UAS-X* (X- *pntP1, pntRNAi, cut, btl, btlRNAi,* or *yan*).

3. To obtain CD8:RFP-marked clones expressing CD4:GFP[11] in random positions of ASP and document the GRASP labeled cytoneme-contacts (*Figure 5D–D''*; *Figure 5—figure supplement 1A–A''*; *Video 4*), *hs-FLP; btl>y+>Gal4; bnl-LexA/TM6* females were crossed to *lexO-*

nsyb:GFP[1-10],UAS-CD4:GFP[11]; UAS-CD8:RFP males. Phalloidin staining was used to mark f-actin enriched cytonemes and the cell cortex.

4. To obtain *bnl* GOF clones in the ASP and check *pntP1* expression (*Figure 6K*): hs-FLP; btl>y+>Gal4,btl-mRFP1moe females were crossed to UAS-bnl:GFP; pntP1-lacZ/TM6 males.

5. To obtain *bnl* GOF clones in the ASP and check *sty* expression (*Figure 6—figure supplement 1D*): hs-FLP; UAS-CD8:GFP; btl>y+>Gal4, btl-mRFP1moe females were crossed to UAS-bnl; sty-lacZ/TM6 males.

6. For *Figure 6L*: *bnl* GOF clones in the ASP were obtained by crossing hs-FLP; btl>y+>Gal4, btl-mRFP1moe females with UAS-bnl:GFP males.

7. To obtain *pntP1* LOF (*Figure 6M*) and GOF (*Figure 6N*) clones in the ASP and observe *cut* expression, hs-FLP; UAS-mCD8GFP; btl>y+>Gal4,btl-mRFP1moe females were crossed to UAS-pntRNAi or UAS-pntP1, respectively.

8. To obtain *cut* GOF clones in the ASP and observe *pntP1* expression (*Figure 6O*), hs-FLP; btl>y+>Gal4,btl-mRFP1moe females were crossed to UAS-cut,UAS-CD8:GFP; pntP1-lacZ/TM6 males.

9. To obtain *cut* LOF clones in the ASP and check *pntP1* expression (*Figure 6P*), hs-FLP; UAS-mCD8GFP; btl>y+>Gal4,btl-mRFP1moe females were crossed to UAS-cutRNAi,pntP1-lacZ/TM6 males.

10. To induce *pntP1* GOF clones in the ASP and check for Btl:Cherry distribution (*Figure 7I*), hs-FLP; btl>y+>Gal4; btl:cherry[endo] females were crossed to UAS-pntP1,UAS-CD8:GFP males.

## Immunohistochemistry

A standard immunostaining protocol was used following (*Du et al., 2017*; *Roy et al., 2014*). To detect the surface exposed/localized Bnl:GFP[ex] or Bnl[ex], standard detergent-free live immunostaining protocol (EIF) was used following (*Schwank et al., 2011*; *Strigini and Cohen, 2000*), except the live tissues were incubated with αBnl or αGFP antibody in optimized WM1 medium (*Du et al., 2017*). Alexa Fluor-conjugated secondary antibodies (1:1000 from Molecular Probes) were used for immunofluorescence detection.

## Generation of anti-Bnl antibody

Affinity-purified rabbit anti-Bnl antibody was generated from YenZym Antibodies, LLC against a peptide containing amino acid residues from 699 to 717 region of the Bnl protein.

## Microscopic imaging

For live imaging, wing imaginal discs and their associated trachea were prepared following (*Roy et al., 2014*). The imaging systems used in this study included Leica SP5X confocal microscope equipped with an ultrasensitive Hybrid Detector, an Andor spinning disc confocal equipped with iXon 897 EMCCD camera, Perkin Elmer UltraVIEW Vox with an EMCCD camera, and a ZeissLSM 800 with Airyscan. Super-resolution images of the Airyscan detector resolved poorly fluorescent and densely populated Bnl:GFP nano-puncta on the ASP/disc cell surface (*Figure 3F–F''*; *Figure 3—figure supplement 2*). The images were processed and analyzed with Fiji. For 3D-rendering, Andor iQ3 software was used. To visualize the faint signals, brightness/intensities of the images were digitally increased following (*Roy et al., 2014*; *Roy et al., 2011b*). For most images, maximum intensity projections of at least 30 optical sections were shown.

## Quantitative analysis of signal gradient

The Intensity profiles of Bnl:GFP, Btl:Cherry, *pntP1*, *cut*, *sty* and *yan* gradients were derived from the maximum intensity projections along the D-P axis of the ASPs. For *Figure 1I–I''*, *Figure 1—figure supplement 2D–E''*, the intensity profiles were derived across the entire digitally straightened epithelium following (*Long et al., 2009*). A ~ 40 pixels wide line was fitted following the curved tracheal/ASP epithelium. For quantification of Bnl:GFP and Bnl:GFP[ex] uptake in mutant clones, total GFP or EIF (red, anti-GFP) intensities were measured from selected areas in mutant clones and neighboring *wt* cells and normalized with the selected area (*Figure 4*; *Figure 4—figure supplement 1*; *Figure 7B,B',E–H*).

## Quantitative analyses of number and orientation of cytonemes

For *Figure 5A–C*, *Figure 7A,A',C–D'*, *Figure 7—figure supplement 1*, cytonemes were manually counted and were grouped by length (<15 μm, 15–30 μm, and >30 μm) as reported in (*Roy et al., 2011b*). Cytonemes emanating from each clone were captured over ~30–40 μm Z-sections with 0.35 μm step size. The R program was used to generate R plots from at least five preparations of each genotype to analyze cytoneme orientation and number.

## Statistical analyses

Statistical significance was determined with a two-tailed t-test (*Figure 3I'; 4E,G'; 7B',E'*), or a one-way ANOVA (*Figure 5B,D''*) followed by Tukey honestly significant different (HSD) test. For all the numerical data analyses, the amount of variation from the mean is indicated by the standard deviation.

## Acknowledgements

We thank Dr. N Andrews, Dr. TB Kornberg and Dr. L Pick for reading and comments on the manuscript; Akshay Patel, Amy Zhou for technical help; Dr. F Port, Dr. K O'Connor-Giles, and Dr. S Feng for discussions on CRISPR strategy; Dr. TB Kornberg, Dr. M Affolter, Dr. M Krasnow, Dr. A Nakamura, Dr. H Bellen, the Bloomington Stock Center, and the Developmental Studies Hybridoma Bank for reagents; Dr. AE Beaven for the UMD imaging core facility; and funding from NIH: R00HL114867 and R35GM124878 to SR.

## Additional information

### Funding

| Funder | Grant reference number | Author |
| --- | --- | --- |
| National Institute of General Medical Sciences | R35GM124878 | Sougata Roy |
| National Heart, Lung, and Blood Institute | R00HL114867 | Sougata Roy |

The funders had no role in study design, data collection and interpretation, or the decision to submit the work for publication.

### Author contributions

Lijuan Du, Conceptualization, Data curation, Formal analysis, Validation, Investigation, Visualization, Methodology, Writing—original draft, Writing—review and editing; Alex Sohr, Formal analysis, Investigation, Visualization, Methodology, Writing—review and editing; Ge Yan, Software, Formal analysis, Methodology; Sougata Roy, Conceptualization, Resources, Data curation, Formal analysis, Supervision, Funding acquisition, Validation, Investigation, Methodology, Writing—original draft, Writing—review and editing

### Author ORCIDs

Lijuan Du (iD) http://orcid.org/0000-0001-5335-8872
Sougata Roy (iD) http://orcid.org/0000-0002-2236-9277

### Decision letter and Author response

Decision letter https://doi.org/10.7554/eLife.38137.058
Author response https://doi.org/10.7554/eLife.38137.059

## Additional files

### Supplementary files

• Source code 1. Source code for the rose plots shown in *Figure 5*, *Figure 7*, and *Figure 7—figure supplement 1*.
DOI: https://doi.org/10.7554/eLife.38137.055

• Transparent reporting form
DOI: https://doi.org/10.7554/eLife.38137.056

### Data availability

All data generated and analysed during this study are included in the manuscript and supporting files. Source data files have been provided for Figure 1,2,3,4,5,6,7 and also for corresponding figure supplements wherever applicable. The source code for R plots in Figures 5 and 7 are provided.

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
