## [Decision Letter]

Thank you for submitting your article "Feedback regulation of cytoneme-mediated transport self-generates a tissue-specific FGF morphogen gradient" for consideration by *eLife*. Your article has been reviewed by two peer reviewers, including Bruce Edgar as the Reviewing Editor and Reviewer #1, and the evaluation has been overseen by Naama Barkai as the Senior Editor.

The reviewers have discussed the reviews with one another and the Reviewing Editor has drafted this decision to help you prepare a revised submission. The reviews in full are appended below. While it is not necessary to address all these comments, many are simply editorial and should be straightforward to address. Indeed both reviewers felt that the manuscript and figures were more complex, and more difficult to read than necessary, so you should spend some time simplifying it according to their suggestions. In addition, reviewer #2 felt that it was quite important that you state the ambivalences concerning the Bnl ligand solubility, and noted that the data you present suggest direct ligand uptake from producer cell membranes, rather than uptake from freely dispersed molecules within the extracellular matrix. GRASP experiments using cell membrane markers from both producer and receiving cells could further clarify this issue, and would be a strong addition to the paper.

*Reviewer #1:*

In this paper, Du et al. describe inter-organ signaling between the *Drosophila* wing disc and the air-sac, a tracheal tissue. They generate flies with GFP and RFP tagged Bnl and Btl, FGF and FGFR homologs used in tracheal morphogenesis, and use these to visualize how Bnl from the disc signals to Btl in the air sac via cytonemes from the air sac cells. The tagged proteins prove to be very informative, and careful imaging shows that there is a gradient of Bnl signaling to the airsac that depends on graded cytoneme density from the airsac cells, which in turn depends upon the response of the cells to Bnl, mediated by the transcription factors Cut and PntP1. The imaging in the paper is done at a very high level (cytonemes are small, dynamic, and tough to fix) and is quite convincing, and a requisite number of genetic tests are included to support the gradient model convincingly. The paper is well written and thoroughly referenced, though somewhat complex. A very interesting aspect of the Bnl signaling gradient is that, through the feedback interactions of its components, it is self-reinforcing. I found the paper quite elegant and have only two general comments.

First, even though the paper is well organized and written, with clear logical conclusions, it is nevertheless complex and difficult to digest. This is in part just because the model, despite its elegance, is unavoidably complex, as are the experiments supporting it. Nevertheless I think the authors should strive to simplify and clarify their delivery as much as is possible. More self-explanatory labels in the figures, and any other possible simplifications should help the reader.

Second, regarding the gradient model, it is easy to understand how it maintains itself (more cytonemes give more Bnl:Btl signaling, more signaling causes more cytonemes), but it is not so clear how it is set up in the first place. The authors' Discussion suggests that they believe they understand gradient establishment (during its development) but how this works was not at all clear to me from the tests shown, or from the summary diagram in Figure 7. It seems there may be more to it, for instance pre-exisiting positional information in the air sac, or direct interactions between the air sac cells, that influence the relative activities of Cut and PntP1 and consequently establish the gradient. Alternatively, maybe the initial guiding factor is just physics – cells closer to the Bnl source get more signaling from the start, even before the graded distributions of Cut, PntP1, Btl, and cytonemes arise. Although it is not necessary for the authors to resolve this issue with more data here, I feel they should discuss the issue of gradient formation more adequately, and perhaps indicate what is possible in their summary figure. In addition, if they can't really explain how the gradient initially forms (as opposed to sustaining itself), then they should change their title.

*Reviewer #2:*

The manuscript by Lijuan Du et al. certainly presents innovative concepts to the field of morphogen function and regulation during tissue organization in development. First, it experimentally establishes in *Drosophila* the existence of a concentration gradient of the FGF molecule Bnl, able to determine the differential activation of target genes in a morphogen-like manner and with direct developmental outcomes. Second, it describes the mechanisms that sustain the establishment of the gradient, which include the involvement of cytonemes. These structures have been increasingly gaining strength as a mechanism for the spatially regulated distribution of signalling molecules. Third, it also initiates the study of cytoneme formation by determining a self-founded positive and negative feedback from the pathway targets towards the regulation of Bnl loaded cytonemes, which in turn regulates target activation. In general the manuscript is well written and presents a coherent development of ideas sustained by experimental work. However, there are a number of major points that could help to clarify the manuscript.

1) It is important to clarify if the ligand Bnl is a soluble molecule or not. The authors should state the ambivalences of the published data in the field concerning solubility of the molecule. The present manuscript and the experimental data presented suggest an uptake of ligand directly from producer cell membranes, rather than an uptake from freely dispersed molecules in the extracellular matrix. Also the authors should attempt to clarify their interpretation regarding the different size of the visualised puncta (<≈2μm and <200nm).

2) The use of the GRASP technique mediated by the Snare protein Syb to illuminate contact points also strongly suggests membrane-membrane contact. The authors fail to discuss this point that would go against the stated solubility of the ligand. The GRASP technique has indeed been reported to mark membrane-to-membrane contact points for the reception of morphogen Hedgehog (González-Méndez et al., 2017; Chen et al., 2017).

3) Finally the understanding of this contact mechanisms, signal transfer and transport of the ligand would benefit from additional experimental data regarding whether cytonemes from the producer cells at the wing disc matter at all. 1) Additional imaging of membranes from ligand-producer cells would aid this crucial point. 2) Cytoneme disruption experiments in the wing disc might clarify this point if no change in distribution is observed. 3) An experiment using the Syb-GRASP technique but reversing the components at each side (producer and receptor sides); or equally one where each half of the molecule GFP is attached to a membrane marker (CD4) on both sides would also clarify this point.

4) In general, presentation of figures needs a more logical arrangement to help with their interpretation. The manuscript would really benefit if figures always included a control beside images and data of mutants. Further labelling is also needed, especially in all the supplementary figures where labelling of different antibodies and reporters within the photo panels would really help their interpretation. In addition, most fluorescent intensity profiles presented don't include information about the sample size used.

5) Particularly the section and corresponding figure regarding the analysis of extracellular and intracellular Bnl ligand (Figure 4 and Figure 4—Figure supplements 1-4) is very confusing. Sometimes contradictory data are presented without further discussion (e.g. lack of Bnl signal-GFP during in vivo imaging).

6) Data comparison between the different mutant conditions that authors intend to correlate not always follow the same parameters; e.g. in Figure 4I and K' data for BnlGFP density is presented for LOF of cytoneme cytoskeleton regulators while density of extracellular Bnl versus intracellular Bnl is presented for Btl GOF. Clarity of the work would be improved if same or equivalent parameters are presented.

---

## [Author Response]

The reviewers have discussed the reviews with one another and the Reviewing Editor has drafted this decision to help you prepare a revised submission. The reviews in full are appended below. While it is not necessary to address all these comments, many are simply editorial and should be straightforward to address. Indeed both reviewers felt that the manuscript and figures were more complex, and more difficult to read than necessary, so you should spend some time simplifying it according to their suggestions. In addition, reviewer #2 felt that it was quite important that you state the ambivalences concerning the Bnl ligand solubility, and noted that the data you present suggest direct ligand uptake from producer cell membranes, rather than uptake from freely dispersed molecules within the extracellular matrix. GRASP experiments using cell membrane markers from both producer and receiving cells could further clarify this issue, and would be a strong addition to the paper.

Thank you for these important suggestions! We rewrote most of the parts of the manuscript and simplified the figures and descriptions of the results as suggested by the reviewers.

We agree that contact-dependent Bnl signaling and its regulation is the key to the mechanism of FGF gradient formation by cytonemes. To further strengthen and clarify this point with experiments, we now show selective localization of the native externalized Bnl at the sites where the ASP cytonemes established direct contacts with the source cells (Figure 3J). In this experiment, super-resolution imaging of the source and recipient cells both labeled with membrane-tethered fluorescent proteins provided an unprecedented resolution of the sites of intimate membrane contact. Amazingly, although the source cells cover a large tissue area compared to the 200 nm thin cytonemes, externalized Bnl puncta were selectively enriched only at the cytoneme contact sites. This experiment shows that Bnl is selectively released at the sites of cytoneme contacts.

We chose to demonstrate this point in this manner instead of using the GRASP technique, because this is a relatively unbiased experimental approach that allows us to preserve the natural membrane affinity and composition of the contact sites while answering the main question – whether Bnl is externalized/released from the source only at these contact points. The *syb*-GRASP or membrane-GRASP is an excellent technique to map and quantify the steady-state number of contact sites. But GFP-reconstitution is also an irreversible reaction, which may (or may not) bias the molecular/signaling composition of the cell/cytoneme membranes at the contact sites. In this context, the experimental results shown in Figure 3J is significant and confirms the GRASP results.

We also provided a quantitative analysis showing that the cytonemes that had stable GRASP contacts with the signal source received higher levels of Bnl than the other cytonemes from the same ASP that did not touch the source. In an earlier paper (Roy et al., 2014), GFP-reconstitution technique was used in the ASP system to thoroughly map the signaling contacts established between the ASP cytonemes and the wing disc *bnl* or *dpp* sources. This study also demonstrated that the cytoneme contacts are essential for both Bnl and Dpp signaling. Several genetic conditions, that removed either the cytonemes or the cytoneme contacts without affecting cytoneme formation from the *btl*-expressing tracheal cells, reduced pMAPK signaling in the mutant ASP cells. However, Bnl dispersion through cytonemes was not visualized earlier, which limited our understanding of the mechanism of Bnl transport and gradient formation. Now, collectively, the results from previous and current work provide substantial evidence that the Bnl exchange is contact-dependent.

We elaborated on the contact-dependence of Bnl signal exchange in our revised Discussion.

Reviewer #1:[…] I found the paper quite elegant and have only two general comments.First, even though the paper is well organized and written, with clear logical conclusions, it is nevertheless complex and difficult to digest. This is in part just because the model, despite its elegance, is unavoidably complex, as are the experiments supporting it. Nevertheless I think the authors should strive to simplify and clarify their delivery as much as is possible. More self-explanatory labels in the figures, and any other possible simplifications should help the reader.

Thank you for the suggestion! In the revised manuscript, we tried our best to simplify the figures, added self-explanatory labels, and clarified the description of the experimental results as much as we could.

Second, regarding the gradient model, it is easy to understand how it maintains itself (more cytonemes give more Bnl:Btl signaling, more signaling causes more cytonemes), but it is not so clear how it is set up in the first place. The authors' Discussion suggests that they believe they understand gradient establishment (during its development) but how this works was not at all clear to me from the tests shown, or from the summary diagram in Figure 7. It seems there may be more to it, for instance pre-exisiting positional information in the air sac, or direct interactions between the air sac cells, that influence the relative activities of Cut and PntP1 and consequently establish the gradient. Alternatively, maybe the initial guiding factor is just physics – cells closer to the Bnl source get more signaling from the start, even before the graded distributions of Cut, PntP1, Btl, and cytonemes arise. Although it is not necessary for the authors to resolve this issue with more data here, I feel they should discuss the issue of gradient formation more adequately, and perhaps indicate what is possible in their summary figure. In addition, if they can't really explain how the gradient initially forms (as opposed to sustaining itself), then they should change their title.

Thank you for this important point. We provided more explanation in the Discussion and in the summary figure (Figure 8). Previously, it was known that the Bnl source remains close to the growing tip of a tracheal branch, and the source changes its position concomitantly with the growing tip. An earlier paper (Du et al., 2017) showed that Bnl sources in the embryo co-migrate in synchrony with the growing tracheal tips. Based on this prior knowledge, we predicted that it is the close association of the source and the recipient cells that provides the initial positional information.

In brief, the feedback mechanisms that we found to regulate the cytonemes operate within a single cell, autonomously. We predict that this mechanism provides a modular control and an interesting mechanism of gradient formation in coordination with the growth and patterning of the ASP. At an early stage prior to ASP budding, a few cells in a preexisting tracheal branch (the disc-associated TC) that are juxtaposed to the Bnl-source could establish cytoneme-contacts with the signal source to induce signaling feedback (PntP1). The TC cells at increasing distance from the source will upregulate Cut or Yan. Thus, the signal and signaling gradients we observed in the ASP could be formed, but their shapes and dimensions are very different and TC specific. With ASP budding and increase in cell number in the growing ASP, as well as with movement of the *bnl-*source, the dimensions of the signal and signaling gradient change. The gradient shape becomes more ASP-specific.

This mechanism implies an interesting systemic feedback where the gradient shape is a result of growth and patterning of the recipient tissue, which it controls. Following the same line of thinking, although it is apparent that an initial physical association is required to initiate the signaling feedback and the gradient, our data might also imply for a self-sustaining feedback between the physical contacts and chemical signaling. Physical contacts of cytonemes are needed for exchange of the chemical signals, whereas the chemical signaling increases the physical connections through feedback on cytoneme formation.

Reviewer #2:[…] In general the manuscript is well written and presents a coherent development of ideas sustained by experimental work. However, there are a number of major points that could help to clarify the manuscript.1) It is important to clarify if the ligand Bnl is a soluble molecule or not. The authors should state the ambivalences of the published data in the field concerning solubility of the molecule. The present manuscript and the experimental data presented suggest an uptake of ligand directly from producer cell membranes, rather than an uptake from freely dispersed molecules in the extracellular matrix. Also the authors should attempt to clarify their interpretation regarding the different size of the visualised puncta (<≈2μm and <200nm).

Thank you for these important suggestions! We emphasized this point in Results and Discussion. FGF proteins are considered to be biochemically soluble/diffusible. However, all our experiments showed that Bnl exchange is contact-dependent, and we characterized how a contact-dependent direct communication is essential for creating a long-range gradient.

Sorry for the confusion! The interpretations of different sized puncta visualized by various imaging methods are now clarified.

2) The use of the GRASP technique mediated by the Snare protein Syb to illuminate contact points also strongly suggests membrane-membrane contact. The authors fail to discuss this point that would go against the stated solubility of the ligand. The GRASP technique has indeed been reported to mark membrane-to-membrane contact points for the reception of morphogen Hedgehog (González-Méndez et al., 2017; Chen et al., 2017).

We added discussion on this important point.

3) Finally the understanding of this contact mechanisms, signal transfer and transport of the ligand would benefit from additional experimental data regarding whether cytonemes from the producer cells at the wing disc matter at all. 1) Additional imaging of membranes from ligand-producer cells would aid this crucial point. 2) Cytoneme disruption experiments in the wing disc might clarify this point if no change in distribution is observed. 3) An experiment using the Syb-GRASP technique but reversing the components at each side (producer and receptor sides); or equally one where each half of the molecule GFP is attached to a membrane marker (CD4) on both sides would also clarify this point.

Thank you for this suggestion! As described earlier, we added new experimental results (new Figure 3J) that show selective localization of Bnl at the sites where ASP cytonemes contact the source cells. We also showed that the cytonemes from the ASP that contacted the source received significantly more signal than the ones that did not contact the source. These experiments, along with previous results published in Roy et al., 2014, provide clear evidence that Bnl exchange is contact-dependent.

For comments 1) and 2): the focus of our study was to characterize the mechanisms for formation of the shapes of the Bnl morphogen gradient, which is recipient tissue-specific, dynamic and adaptive. Our source and recipient membrane-marked images showed that the ASP cytonemes directly contact the source cell surface and exchange the signal. We could not visualize the source cytonemes under the imaging conditions we used. Therefore, we did not perform the experiments involving removal of the source cytonemes. We would like to investigate the possibility of source cytonemes and their roles in the future. We added a discussion on cytoneme-mediated signaling mechanisms.

Although we did not perform this experiment, all our experiments suggest that signaling through a gradient of number of cytonemes across the recipient ASP epithelium and their spatial regulation by two counteracting feedback of Bnl signaling generate recipient tissue-specific shapes of the morphogen gradient. In this study, we identified only a small set of molecules and events that might regulate these core feedback processes. We anticipate additional unknown cellular and molecular events that feed into the central feedback loops. Unknown mechanisms might be involved in guiding the recipient cytonemes to find the target and establish contact with the source, assisting in signal exchange and transport via motor proteins, or endocytosing/exocytosing the signal. An example of such cellular and molecular event could be the source cytonemes, which might deliver Bnl to the recipient cytonemes. According to this hypothesis, a mechanism of cytoneme-mediated signal delivery directly to the recipient cytonemes would contribute to the levels of Bnl in recipient ASP cells, and thereby to the feedback processes inducing/suppressing cytoneme formation and signal uptake. A loss of source cytonemes would jeopardize the whole feedback response mechanism that creates the gradient, and is expected to yield an effect similar to the *bnl* knockdown experiment, leading to a simultaneous loss of both the ASP-specific gradient and the ASP development (Figure 1—figure supplement 2B-C). We would like to test this possibility in the future.

4) In general, presentation of figures needs a more logical arrangement to help with their interpretation. The manuscript would really benefit if figures always included a control beside images and data of mutants. Further labelling is also needed, especially in all the supplementary figures where labelling of different antibodies and reporters within the photo panels would really help their interpretation. In addition, most fluorescent intensity profiles presented don't include information about the sample size used.

Thank you for the suggestion! We reorganized the figures, included controls whenever necessary, and added more labeling to facilitate figure interpretation. Also, we added information about the sample size whenever applicable.

*5) Particularly the section and corresponding figure regarding the analysis of extracellular and intracellular Bnl ligand (Figure 4 and Figure 4—figure supplements 1-4) is very confusing. Sometimes contradictory data are presented without further discussion (e.g. lack of Bnl signal-GFP during* in vivo *imaging).*

Sorry for the confusion and thanks for this helpful comment! We now simplified the text and clarified all these points in the Results and Discussion. We reorganized the figures in this section, and added one figure panel (new Figure 3—figure supplement 2B, B’) to further clarify the externalized and intracellular Bnl ligands. Note that the original Figure 4 is now Figure 3 in the revised manuscript.

Bnl:GFP is expressed at a low level from the endogenous locus. To detect Bnl:GFP molecules, we imaged Bnl:GFP distribution using various technologies such as super-resolution imaging as well as detergent-free immunostaining (EIF) to identify the population of post-secretion Bnl:GFP. Using standard confocal imaging in combination with the EIF assay, we found that a large amount of Bnl:GFP is localized on the surface of the ASP and on the ASP cytonemes. However, the externalized molecules present on the cytonemes were hard to detect only with GFP fluorescence due to the low levels of the fluorescence emitted from endogenous signal. With a high-gain super-resolution imaging we overcame this technical challenge. All experimental results supported cytoneme-mediated transport.

6) Data comparison between the different mutant conditions that authors intend to correlate not always follow the same parameters; e.g. in Figure 4I and K' data for BnlGFP density is presented for LOF of cytoneme cytoskeleton regulators while density of extracellular Bnl versus intracellular Bnl is presented for Btl GOF. Clarity of the work would be improved if same or equivalent parameters are presented.

Thanks for the suggestion! We reorganized the related figures (new Figure 4) to make them easier to follow. Same equivalent parameters were used to measure in the clonal analyses. We now explained the graphs thoroughly in the text (now Figure 4E).

Figure 4K’ (now Figure 4G’): We aimed to display the autonomous loss of the surface-localized Bnl:GFP (Bnl:GFP^ex^) observed in receptor-LOF clones. In receptor-LOF clones, intracellular levels of Bnl:GFP is expected to get reduced irrespective of how Bnl:GFP travels in extracellular space. Since, most Bnl:GFP puncta in the ASP detected under standard confocal imaging were representing intracellular endocytosed signal (Figures 2 and 3), we did not combine the *btl-*LOF data with cytoneme-regulators. However, an EIF-stained Bnl:GFP^ex^ levels on the LOF clones could clearly distinguish between the cytoneme-mediated dispersion versus a passive diffusion mechanism (Figure 4F-G). Therefore, we compared the levels of Bnl;GFP^ex^ inside and outside of the mutant clones. We now explained the results and simplified the figure and relevant texts to clarify this point.